# POISSON PROCESS FOR BAYESIAN OPTIMIZATION

## ABSTRACT

Bayesian Optimization (BO) is a sample-efficient, model-based method for optimizing black-box functions which can be expensive to evaluate. Traditionally, BO seeks a probabilistic surrogate model, such as Tree-structured Parzen Estimator (TPE), Sequential Model Algorithm Configuration (SMAC), and Gaussian process (GP), based on the exact observed values. However, compared to the value response, relative ranking is harder to be disrupted due to noise resulting in better robustness. Moreover, it has better practicality when the exact value responses are intractable, but information about candidate preferences can be acquired. This work introduces an efficient BO framework, namely Poisson Process Bayesian Optimization (PoPBO), consisting of a novel ranking-based response surface based on Poisson process and two acquisition functions to accommodate the proposed surrogate model. We show empirically that PoPBO improves efficacy and efficiency on both simulated and real-world benchmarks, including HPO and NAS. Source code will be made publicly available.

## 1 INTRODUCTION

Bayesian optimization (BO) (Mockus et al., 1978) is a popular black-box optimization paradigm and has achieved great success in a number of challenging fields, such as robotic control (Calandra et al., 2016), biology (González et al., 2015), and hyperparameter tuning for complex learning tasks (Bergstra et al., 2011). A standard BO routine usually consists of two steps: (1) Learning a probabilistic response surface that captures the distribution of an unknown function $f(x)$; (2) Optimizing an acquisition function that suggests the most valuable points for the next query iteration. Popular response surface for the first step includes Random Forest (SMAC) (Hutter et al., 2011), Tree-structure Parzen Estimator (TPE) (Bergstra et al., 2011), Gaussian Process (GP) (Snoek et al., 2012) and Bayesian Neural Network (BNN) (Springenberg et al., 2016; Snoek et al., 2015). Acquisition functions for the second step include Expected Improvement (EI) (Mockus, 1994), Thompson Sampling (TS) (Chapelle & Li, 2011; Agrawal & Goyal, 2013) and Upper/Lower Confidence Bound (UCB/LCB) (Srinivas et al., 2012), which are designed to trade off exploration and exploitation.

Most of the existing BO methods (Bergstra et al., 2011; Hutter et al., 2011; Snoek et al., 2012) adopt absolute response surfaces [1] that attempt to fit the black-box function based on the observed absolute function values. However, such an absolute metric can have the following disadvantages. **1) Absolute response can be difficult to obtain or even unavailable in some practical scenarios**, such as sports games and recommender systems where only relative evaluation [2] can be provided by pairwise comparison (He et al., 2022). **2) Absolute response can be sensitive to noise**, which is also pointed out by Rosset et al. (2005). Such an issue will affect the performance of BO in real-world scenarios, where absolute responses are usually noisy. **3) It can be challenging to directly transfer absolute response surfaces.** In particular, multi-fidelity metrics usually have different absolute responses for the same candidate, making it hard to utilize history observations on a coarse-fidelity metric to warm up the training of surrogate models on a fine-grained-fidelity one. Similarly, in hyperparameter optimization (HPO) and neural architecture search (NAS) tasks, performance on different datasets of the same hyperparameter selection or neural architectures is also different and is hard to be transferred across datasets.

---

[1] In this work, 'absolute evaluation (response)' of one query is defined as its exact black-box function value.

[2] In this work, 'relative evaluation (response)' of one query is defined as its ranking, which can be computed by comparing with other candidates.

Relative metrics can be an effective cure for the above issues. **1) Relative response such as ranking has better practicality** when the information about candidate preferences can be more easily acquired than raw value (González et al., 2017), which is also widely used in many prior works (Kahneman & Tversky, 2013; Brusilovsky et al., 2007; González et al., 2017). **2) Relative response is more robust to noise than absolute response** since relation such as ranking between candidates is harder to be disrupted to noise, but absolute values are sensitive. In this work, we analyze the robustness of rankings in Sec. 3.1 under the common additive Gaussian noise assumption, showing that rankings are more insensitive to noise than absolute values. Similar conclusions are also made in other areas regarding the advantage of ranking models e.g. (Rosset et al., 2005). **3) Relative response has better transferability**, such as rankings between candidates, since they are usually comparable among multi-fidelity metrics or evaluations across different datasets for the same candidate. It is also demonstrated by (Salinas et al., 2020; Nguyen et al., 2021; Feurer et al., 2018).

Some Bayesian Optimization methods also adopt relative responses and are related to our work. Preferential BO methods (González et al., 2017; Mikkola et al., 2020) attempt to capture the relative preference by comparing pair of candidates. However, they have to rely on a computational-expensive soft-Copeland score (PBO) or have to optimize EI by the projective preferential query (PPBO) to propose the next query (optimal candidate). Moreover, they ignore tie situations, which commonly exist in real scenarios. Nguyen et al. (2021) extend the above method by comparing k samples. Specifically, they utilize Gaussian Process to model the absolute function values and leverage a multi-nominal logit model to build the evidence likelihood of local ranking of k observations. Although this method overcomes the computational disadvantage and takes ties into account, it essentially models the absolute response and simply captures the relationship (local ranking) among k candidates.

In contrast to the above methods, we propose to capture the global ranking of each candidate in a feasible domain (search space) and model the relative response. On the one hand, we can directly search the optimum based on our relative response surface and obtain the next query without computationally expensive procedure (González et al., 2017; Mikkola et al., 2020). On the other hand, unlike (Nguyen et al., 2021) that first build an absolute response surface and then derive local ranking among k candidates as the evidence, our method directly fits a ranking-based relative response surface. Moreover, due to the nature of ranking, our method can handle tie situations where candidates have the same ranking. Specifically, we adopt Poisson Process (PP) to capture the global ranking, which is naturally suitable since the ranking of a candidate can be figured out by counting the number of better candidates. Fig. 1 shows the superiority of our response surface to capture the global ranking against the GP-based one. Specifically, we conduct experiments on the Forrester function with various degrees of additive Gaussian noise. The setting details can be found in Appendix C.1. Our response surface is more robust to noise and can better capture the global ranking. Furthermore, we derive two acquisition functions to accommodate our response surface for a better exploitation-exploration trade-off. Finally, we propose a novel Bayesian Optimization framework, named PoPBO, achieving lower regret (better performance) with faster speed. Our contributions can be summarized as follows:

**1) Ranking-based Response Surface based on Poisson Process.** Unlike the prior absolute response surface (Bergstra et al., 2011; Snoek et al., 2012), nor those (Nguyen et al., 2021) using relative evidence likelihood based on absolute responses, this work, to the best of our knowledge, is the first to directly capture the global ranking over a feasible domain via Poisson process. The robustness against noise is also analyzed in Sec. 3.1 and illustrated in Fig. 1.

**2) Tailored Acquisition Function for Ranking-based Response Surface.** Two acquisition functions for our response surface, named R-LCB and ERI, are deduced from the vanilla LCB and EI for better exploitation-exploration trade-off. Gradients of the proposed acquisition functions w.r.t. candidates are also derived, so the next query can be optimized by SGD.

**3) Computational-Efficient Bayesian Optimization Framework.** The proposed ranking-based response surface and acquisition functions form a novel Bayesian optimization framework: Poisson Process Bayesian Optimization (PoPBO). Our framework is much faster than Gaussian process-based BO methods. Specifically, the computational complexity of PoPBO is $O(N^2)$ compared to $O(N^3)$ of GP, where $N$ is the number of samples (see Fig. 3).

**4) Extensive Empirical Study with Strong Performance.** Our method achieves substantial improvements over many prior BO methods on the simulated functions and multiple benchmarks on real-world datasets, including hyperparameter optimization and neural architecture search.

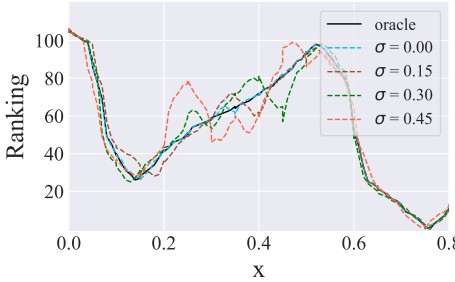 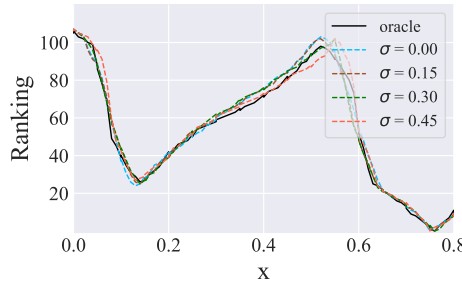

(a) GP (value-based) Response Surface        (b) PP (ranking-based) Response Surface

Figure 1: We compare the sensitivity of additive Gaussian noise between GP (value-based) response surface and PoPBO (ranking-based response surface) on the Forrester function. Based on the Forrester function value, the solid black line (oracle) indicates actual rankings over 100 points evenly spaced from 0 to 0.8. We draw lines between the 100 predictions by linear interpolation for a clear illustration. The dash lines indicate predicted rankings over the 100 points by (a) Gaussian process and (b) Poisson process on observations with varying degrees of noise whose standard deviation $\sigma$ ranges from 0 to 0.45. Each response surface is trained on the same 15 queries. Note that GP performs worse as the standard deviation of noise increases. In contrast, PP performs consistently well due to its great robustness against noise in our analysis.

## 2   PRELIMINARIES AND BACKGROUND

**Bayesian Optimization.** Consider minimizing a black-box target function $f(\cdot)\colon X \to \mathbb{R}$ defined on a d-dimensional feasible domain $X \subset \mathbb{R}^d$, our goal is to find a global minimum of $f(\cdot)$.

$$x^* = \arg\min_{x \in X} f(x). \tag{1}$$

These black-box objective functions can be expensive to be evaluated, such as those without closed-form expression/derivative functions and long-time training neural networks. Bayesian Optimization (BO) is an efficient method to solve these problems (Brochu et al., 2010; Perrone et al., 2018). It relies on a response surface built on the black-box function, which fits sequentially using the new query selected according to an exploit-explore trade-off acquisition function. A Gaussian Process with a nonlinear kernel is one of the most popular choices of the response surface, which places a GP prior to the unknown function $f$. It gives its posterior in an exact form conditioned to the prior observations $D = \{(x_j, y_j)\}_{j=1}^{J}$. SMAC (Hutter et al., 2011) introduces random forest models for regression, which can be used to handle those categorical hyperparameters. TPE (Bergstra et al., 2011) is another BO method that models two densities $l(x) = p(y < \alpha | x, D)$ and $g(x) = p(y > \alpha | x, D)$ via kernel density estimator instead of modeling $p(y|x)$ directly like GP, and then the ratio $l(x)/g(x)$ will be optimized as the acquisition function to find the next query. Recently, Bayesian neural network was introduced into BO framework (Snoek et al., 2015) to model the response surface due to its flexibility and scalability, and Springenberg et al. (2016) using a more robust stochastic gradient MCMC method (Chen et al., 2014) to evaluate the posterior. Besides a powerful response surface, BO also needs an exquisite criterion for suggesting the next query considering the trade-off between exploitation and exploration. The most common acquisition function is EI since it is intuitive and has achieved strong performance in various tasks. Another common criterion is LCB, which minimizes regret during sequential optimization.

**BO with Relative Metrics.** The relative metric does not have to utilize absolute responses of the black-box function. Some methods focus on the cases where the function evaluation is not directly accessible (Brusilovsky et al., 2007; González et al., 2017; Mikkola et al., 2020; Siivola et al., 2021). Absolute responses can be difficult to obtain or even unavailable in some practical scenarios, such as sports games and recommender systems (Brusilovsky et al., 2007) where only relative evaluation can be provided by pairwise comparisons. *Preferential Bayesian Optimization* (PBO) (González et al., 2017) captures correlations between different inputs to find the optimal value of a latent function, which requires limited comparisons. To handle a high-dimensional black-box function, *Projective Preferential Bayesian Optimization* (PPBO) (Mikkola et al., 2020) proposes a projective preferential query allowing for the feedback given by human interaction. However, they ignore the

tie situations and have to rely on a computationally expensive procedure to suggest the next query. Nguyen et al. (2021) extend the above method by comparing k samples but have to model the absolute response surface by Gaussian Process and assume the noise obeys Gumbel distribution. In addition, ranking-based methods (Feurer et al., 2018; Salinas et al., 2020) can also facilitate the identification of similar runs for transfer learning, reusing insights from past similar experiments. This work, on the contrary, makes the first attempt to directly capture the global ranking of candidates based on Poisson process and derive a novel Bayesian Optimization framework named PoPBO. We analyze the robustness of relative metric (ranking) against noise and show the outstanding performance of our method on various simulated benchmarks and real-world datasets.

## 3 POISSON PROCESS FOR BAYESIAN OPTIMIZATION

This section first introduces a ranking-based evaluation and analyzes its robustness compared to the value-based evaluation. Then, we propose a novel response surface based on Poisson process to capture the ranking-based evaluation for each candidate and derive the log-likelihood on observations. Finally, we fit the ranking-based response surface by training the weights through SGD and then provide the posterior probability.

### 3.1 RANKING-BASED EVALUATION

Suppose a black-box score function $f(\cdot)$ is defined on a feasible domain $X$ consisting of all optional candidates. Given a sample $x \in X$ and a subset of the feasible domain $S \subset X$, we define a set $S_x = \{y | y \in S, f(y) < f(x)\}$, consisting of the better candidates than $x$ in a set $S$. Considering the physical meaning of $S_x$, we can estimate the superiority of $x$ for $f(\cdot)$ against the points in set $S$ by measuring $S_x$. Specifically, considering two points $x_1, x_2$, $S_{x_1}$ has a larger measure value, representing that there are more points in $S$ better than $x_1$, so $x_1$ is worse than $x_2$.

To construct such a metric, we first introduce sets $\mathcal{Q} = \{S_x | S \subset X, x \in X\}$. Given a sampled set $\hat{S} \subset X$. We can define a metric $\hat{\mu}$ on the space $(X, \mathcal{Q})$ by the number of elements in $S_x \cap \hat{S}$:

$$\hat{\mu}(S_x) = |S_x \cap \hat{S}|. \tag{2}$$

**Robustness Analysis.** Suppose a black-box function $\hat{f}(x)$ with additive Gaussian noisy observations $f(x) = \hat{f}(x) + \epsilon, \epsilon \sim \mathcal{N}(0, \sigma^2)$. Consider two queries $x_1, x_2$ with observations $f(x_1), f(x_2)$. We assume $\hat{f}(x_1) < \hat{f}(x_2)$ without loss of generality, the probability of correctly ranking $x_1, x_2$ is:

$$P(f(x_1) < f(x_2)) = P(\epsilon_1 - \epsilon_2 < \hat{f}(x_2) - \hat{f}(x_1)). \tag{3}$$

Since $\epsilon_1, \epsilon_2 \sim \mathcal{N}(0, \sigma^2)$ are independent variables, $\Delta\epsilon = \epsilon_1 - \epsilon_2 \sim \mathcal{N}(0, 2\sigma^2)$. According to three-sigma rule of thumb, if $\hat{f}(x_2) - \hat{f}(x_1) > \sqrt{2}\sigma$, the probability of correctly ranking $x_1, x_2$ is larger than 82.63%; If $\hat{f}(x_2) - \hat{f}(x_1) > 2\sqrt{2}\sigma$, the probability of correctly ranking $x_1, x_2$ is larger than 97.72%. Therefore, even if observations are noisy, the ranking of candidates is hard to be disrupted. Fig. 1(b) verifies the robustness to noise of ranking-based evaluation. Specifically, we do not involve any prior distribution for noise when training our PP response surface, but it can still capture the correct ranking (black line) under various noise levels.

### 3.2 CAPTURING THE RANKING VIA POISSON PROCESS

Given a sample $x$ and a set $\hat{S}$, we utilize a random process $\hat{R}_x(S), \forall S \subset X$ to capture the ranking $x$. Though $\hat{R}_x(S)$ also depends on $\hat{S}$, we omit it for conciseness. In particular, we define $\hat{R}_x \triangleq \hat{R}_x(X)$ to denote the ranking of $x$ over the whole feasible domain $X$, which is a random variable. We can model $\hat{R}_x(S), \forall S \subset X$ as independent increment counting process, since the objective function $f(x)$ is black-box and we assume the rankings of $x$ over two disjoint areas are independent, i.e., $\hat{R}_x(S_1) \perp\!\!\!\perp \hat{R}_x(S_2), \forall S_1, S_2 \subset X, S_1 \cap S_2 = \varnothing$. Moreover, $\hat{R}_x(S)$ has the following properties: 1) $\hat{R}_x(\varnothing) = 0$ and 2) $\lim_{\Delta s \to 0} P(\hat{R}_x(S + \Delta s) - \hat{R}_x(S) \geq 2) = 0, \forall S \subset X$. Detailed discussion is provided in Appendix A. Since the supremum of $\hat{R}_x(S)$ is $|\hat{S}|$, $\hat{R}_x(S)$ obeys truncated non-homogeneous Poisson process (Yigiter & Inal, 2006) as Eq. 4 with parameter $\lambda(s, x), s \in X$.

$$\hat{R}_x(S) \sim Poisson\left(\int_S \lambda(s, x) \mathrm{d}s\right). \tag{4}$$

Hence, the ranking of $x$ over the whole feasible domain is $\hat{R}_x = \hat{R}_x(X)$, the probability of $\hat{R}_x = k$ is:

$$\mathrm{P}\left(\hat{R}_x = k|x, \hat{S}\right) = \frac{\left(\int_X \lambda(s, x)\mathrm{d}s\right)^k}{k! \cdot Z(x)} \exp\left(-\int_X \lambda(s, x)\mathrm{d}s\right) = \frac{(\lambda_\xi(x)|X|)^k}{k! \cdot Z(x)} \exp\left(-\lambda_\xi(x)|X|\right), \quad (5)$$

$$Z(x) = \sum_{k=0}^{|\hat{S}\backslash\{x\}|} \left[\frac{(\lambda(\xi, x)|X|)^k}{k!} \exp\left(-\lambda(\xi, x)|X|\right)\right], \quad (6)$$

where $Z(x)$ is the normalized coefficient and $|\hat{S}\backslash\{x\}|$ indicates the number of sampled sets without $x$. There exists $\xi \in X$ satisfying $\int_X \lambda(s, x)\mathrm{d}s = \lambda_\xi(x)|X|$ according to the mean value theorem for integrals. We can approximate $\lambda_\xi(x)$ by a multi-layer perception (MLP) $\lambda_\xi(x; \theta)$ with parameter $\theta$.

Give $N(N >= 2)$ samples $\hat{S} = \{x_j\}_{j=1}^N$, the ranking of each sample over $\hat{S}$ is $\hat{K} = \{\hat{k}_{x_j}\}_{j=1}^N$. With the independent assumption of observations similar to (Salinas et al., 2020), the log-likelihood is:

$$\log L(\hat{K}|\hat{S}; \theta) = \sum_{j=1}^N \left\{\hat{k}_{x_j} \log\left(\lambda_\xi(x_j; \theta)|X|\right) - \log(\hat{k}_{x_j}!) - \log\left[\sum_{i=0}^{N-1} \frac{(\lambda_\xi(x_j; \theta)|X|)^i}{i!}\right]\right\}. \quad (7)$$

We can train the weights $\theta$ by minimizing the negative log-likelihood on observations in Eq. 7 by SGD, whose gradient can be computed as follows:

$$\frac{\partial}{\partial\theta}\left(-\log L(\hat{K}|\hat{S})\right) = \sum_{j=1}^N \frac{\partial\lambda_\xi(x_j; \theta)}{\partial\theta} \cdot \left[\frac{\hat{k}_{x_j}}{\lambda_\xi(x_j; \theta)} - \frac{\sum_{i=0}^{N-2} \frac{(\lambda_\xi(x_j; \theta)|X|)^i}{i!}|X|}{\sum_{i=0}^{N-1} \frac{(\lambda_\xi(x_j; \theta)|X|)^i}{i!}}\right]. \quad (8)$$

Once $\theta$ is determined after training on the observations $(\hat{S}, \hat{K})$, the ranking of a new sample $x^*$ over the whole feasible domain $X$ can be predicted, where $Z$ is the normalized coefficient by Eq. 6.

$$\mathrm{P}\left(\hat{R}_{x^*}(X) = k|\theta, x^*, \hat{S}\right) = \frac{\left(\lambda_\xi(x^*; \theta)|X|\right)^k}{k! \cdot Z(x^*)} \cdot \exp\left(-\lambda_\xi(x^*; \theta)|X|\right). \quad (9)$$

The proposed Bayesian optimization framework with Poisson process (PoPBO) is outlined in Alg. 1 in Appendix B. The acquisition function will be introduced in the next section.

## 4 ACQUISITION FUNCTION FOR POPBO

The existing acquisition functions are designed for absolute response surface considering independent mean and variance, which can be improper for our response surface since the mean of Poisson distribution is the same as the variance. Directly applying these acquisition functions to our PoPBO will cause the over-exploitation issue. To this end, we introduce a series of acquisition functions, named Rectified Upper Confidence Bound (R-LCB) and Expected Ranking Improvement (ERI), derived from vanilla LCB and EI, respectively.

### 4.1 RECTIFIED LOWER CONFIDENCE BOUND (R-LCB)

The ranking of each point $x$ obeys Poisson distribution as Eq. 9, with expectation $\mu(x) = \lambda_\xi(x)|X| \cdot \frac{\sum_{i=0}^{N-1}(\lambda_\xi(x_j; \theta)|X|)^i/i!}{\sum_{i=0}^{N}(\lambda_\xi(x_j; \theta)|X|)^i/i!}$ and standard deviation $\sigma(x) = \sqrt{\mu(x)}$. Thus the vanilla LCB of each point is:

$$\alpha_{\mathrm{LCB}}(x) = \mu(x) - \beta\sigma(x) = \sqrt{\mu(x)}\left(\sqrt{\mu(x)} - \beta\right). \quad (10)$$

However, Poisson distribution with a large expectation has a larger variance, indicating less confidence in the ranking prediction. So the vanilla LCB will be easily trapped into an over-exploitation issue. Therefore, we propose to restrict the lower value as a threshold and define the rectified LCB (R-LCB):

$$\alpha_{\mathrm{R\text{-}LCB}}(x) = \begin{cases} \alpha_{\mathrm{LCB}}(x) & \text{if } \lambda_\xi(x; \theta)|X| < q|\hat{S}| \\ \epsilon_x, \epsilon_x \sim U[0, 1] & \text{Otherwise} \end{cases}, \quad (11)$$

where $\epsilon_x$ is used for re-parameterization, and $q$ is some quantile of the number of existing samples, according to which, the threshold $q|\hat{S}|$ can be adaptively adjusted during the BO process. To minimize R-LCB, we randomly sample a set of start points and adopt LBFGS (Liu & Nocedal, 1989) for optimization. In particular, LBFGS will not update the samples whose predicted ranking is larger than $q|\hat{S}|$, and they have a probability of being selected as the next query if the sampled $\epsilon_x$ is very small. We set $q = 0.6$ by default. Results in Fig. 6 show the advantage of our R-LCB against LCB.

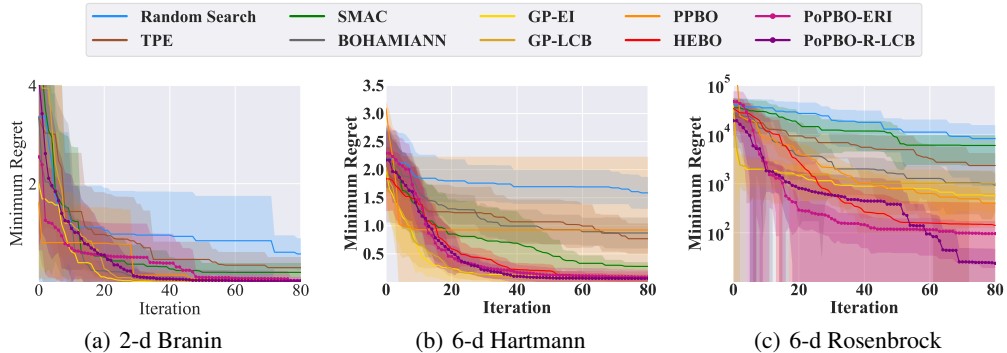

Figure 2: Performance of various black-box optimization methods on three simulation functions. Y-axis indicates the residual between the optimum function value and the incumbent. We run each method ten times and plot the average performance and standard deviation as the line and shadow.

## 4.2 EXPECTED RANKING IMPROVEMENT (ERI)

Inspired by EI, we introduce ERI to maximize the expected improvement on ranking over the worst tolerable ranking $K_m$ to balance exploitation and exploration. We set $K_m = 5$ by default.

$$\alpha_{\text{ERI}}(x) = \sum_{k=0}^{K_m} (K_m - k) \cdot \text{P}\left(\hat{R}_x = k | \theta, x\right), \tag{12}$$

where $\text{P}\left(\hat{R}_x = k | \theta, x\right)$ is defined in Eq. 9 representing the prediction of ranking of $x$. The gradient of ERI w.r.t. $x$ is defined as follows, where $\lambda_\xi(x) = \lambda_\xi(x; \theta)$.

$$\frac{\partial \alpha_{\text{ERI}}(x)}{\partial x} = \sum_{k=0}^{K_m} \left[ \frac{K_m - k}{k!} \frac{\partial}{\partial x} \left( \frac{(\lambda_\xi(x)|X|)^k}{k! \cdot Z(x)} \cdot \exp\left(-\lambda_\xi(x)|X|\right) \right) \right]$$

$$= \sum_{k=0}^{K_m} \left\{ \frac{(K_m - k)(\lambda_\xi(x)|X|)^{k-1}|X|}{k!\left(\sum_{i=0}^{N} \frac{(\lambda_\xi(x)|X|)^2}{i!}\right)^2} \left[ k \sum_{i=0}^{N} \frac{(\lambda_\xi(x)|X|)^2}{i!} - \lambda_\xi(x)|X| \sum_{i=0}^{N-1} \frac{(\lambda_\xi(x)|X|)^2}{i!} \right] \right\}.$$

Hence, we can get the next query $x^*$ by minimizing $\alpha_{\text{ERI}}(x)$ through LBFGS optimizer. Similar to R-LCB, we also apply the rectified technique in Eq. 11 to ERI.

## 5 EMPIRICAL ANALYSIS

**Benchmarks.** We verify the efficacy of PoPBO on both simulated and real-world benchmarks, including HPO and NAS. For the simulated benchmark, we apply PoPBO to optimize three simulation functions: 1) 2-d Branin function with the domains of each dimension are $[-5, 10]$ and $[0, 15]$ respectively; 2) 6-d Hartmann function in $[0, 1]$ for all six dimensions; 3) 6-d Rosenbrock function defined in $[-5, 10]^6$. For the HPO task, we test PoPBO on the tabular benchmark HPO-Bench (Eggensperger et al., 2021), containing the root mean square error (RMSE) of a 2-layer feed-forward neural network (FCNET) (Klein & Hutter, 2019) trained under 62208 hyper-parameter configurations on four real-world datasets: protein structure (Rana, 2013), slice localization (Graf et al., 2011), naval propulsion (Coraddu et al., 2016) and parkinsons telemonitoring (Tsanas et al., 2010). The averaged RMSE over four independent runs under the same configuration is utilized as the performance of that configuration. For the NAS task, we test on NAS-Bench-201 (Dong & Yang, 2020) containing 15,625 architectures in a cell search space that consists of 6 categorical parameters, and each parameter has five choices. Each architecture is evaluated on three datasets. Following the setting of (Dong & Yang, 2020), we search for the best architecture according to its performance on the CIFAR-10 validation set after 12 epochs training.

**Baselines.** We compare against random search (RS) (Bergstra & Bengio, 2012) and various value-based Bayesian optimization methods, including Gaussian Process (GP) (Snoek et al., 2012), Tree-structured Parzen Estimation (TPE) (Bergstra et al., 2011), and Sequential Model Algorithm Configuration (SMAC) (Hutter et al., 2011), BOHAMIANN (Springenberg et al., 2016), and HEBO (Cowen-Rivers et al., 2020). For GP methods, we use EI and LCB as acquisition functions, which are

optimized by LBFGS, and adopt the ARD Matérn 5/2 covariance function to be the kernel function. We also compare with PPBO (Mikkola et al., 2020), one of state of the art preferential BO methods that also utilizes relative response (preferential between pair of candidates). Detailed settings of the baselines are provided in Appendix C.2.

**Settings.** We run all the methods for 80 iterations with 12 initial points by default. For the Rosenbrock-6d simulation function, we run all methods for 80 iterations with 30 initial points due to its complex search space. The MLP $\lambda_\xi(x; \theta)$ used to approximate the parameter of the Poisson process has three hidden layers with 128 nodes and a ReLU activation function. The MLP is trained for 100 steps by ADAM with 64 batch sizes and a 0.01 initial learning rate multiplied by 0.2 every 30 steps. All methods are evaluated ten times independently on an Intel(R) Xeon(R) Silver 4210R CPU.

## 5.1 PERFORMANCE ON THE SIMULATED BENCHMARKS

**2-d Branin.** Fig. 2(a) compares PoPBO and baselines on Branin, which is a widely used simulation benchmark, verifying the efficacy of our PoPBO with both ERI and R-LCB.

**6-d Hartmann.** This function has higher dimensions than Branin and thus is more difficult to optimize. As shown in Fig. 2(b), although the standard GP temporarily outperforms others in the early stage, our PoPBO achieves the best at around 40-th iterations and takes the lead till the end of the BO procedure. In particular, PPBO performs great in the early stage but falls into a local optimum after ten iterations.

**6-d Rosenbrock.** Optimizing the 6-d Rosebrock function is much more complicated than Branin and Hartmann since its global optimum lies in a narrow valley (Picheny et al., 2013) as well as a more extensive search space. Hence, we increase the initial points to 30 for a better preview of the Rosenbrock landscape for all methods. Fig. 2(c) shows that our PoPBO can quickly find the valley and significantly outperforms other BO methods.

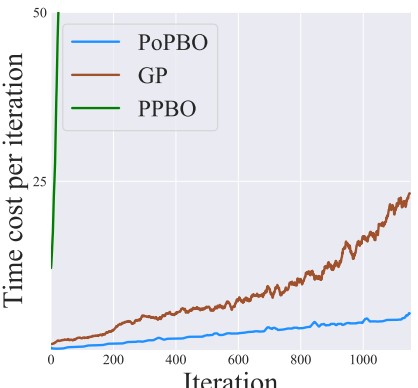

Figure 3: Time cost of Gaussian process Bayesian optimization (GP), PPBO (Mikkola et al., 2020) and PoPBO. All the methods are applied to optimize 6-d Hartmann function.

**Computational Cost.** Fig. 3 compares the time cost of three peer methods, showing that the cost of GP-BO and PPBO are much higher than PoPBO as the number of observations increases. Specifically, GP has to compute the inverse of a covariance matrix resulting in a $O(N^3)$ computational complexity. PPBO is also based on GP and requires computing another covariance matrix of size $J \times J$, where $J$ is the number of random samples during optimization. In contrast, the computational bottleneck of PoPBO lies in the training of an MLP, which is $O(N^2)$ as shown in Eq. 7.

## 5.2 PERFORMANCE ON THE REAL-WORLD BENCHMARKS

**HPO-Bench.** we run each method for ten times and plot the trend of minimum regret during the BO procedure. Fig. 4 compares PoPBO with advanced Bayesian optimization methods and random search on HPO-Bench (Eggensperger et al., 2021), showing that our PoPBO achieves the best on all the four datasets. In contrast, other methods are unable to perform consistently well and even worse than random search. Moreover, the performance of our method has a lower standard deviation than other methods, indicating its outstanding stability. The numerical performance of all methods on the four datasets are provided in Table 2 in Appendix C.3.

**NAS-Bench-201.** Table 1 reports the performance on the NAS task. The first block shows the performance of prior non-parameter-sharing-based NAS methods, including random search, evolution algorithm, reinforcement learning, and Bayesian optimization. We adopt the same initial observations when testing Random Search, GP-BO, and our PoPBO for a fair comparison. As for SMAC and TPE, we directly run the open-source codes, which have different sampling implementations from ours, making it hard to sample the same initial observations as ours even under the same random seed. Our method achieves the best performance on the three datasets and, in particular,

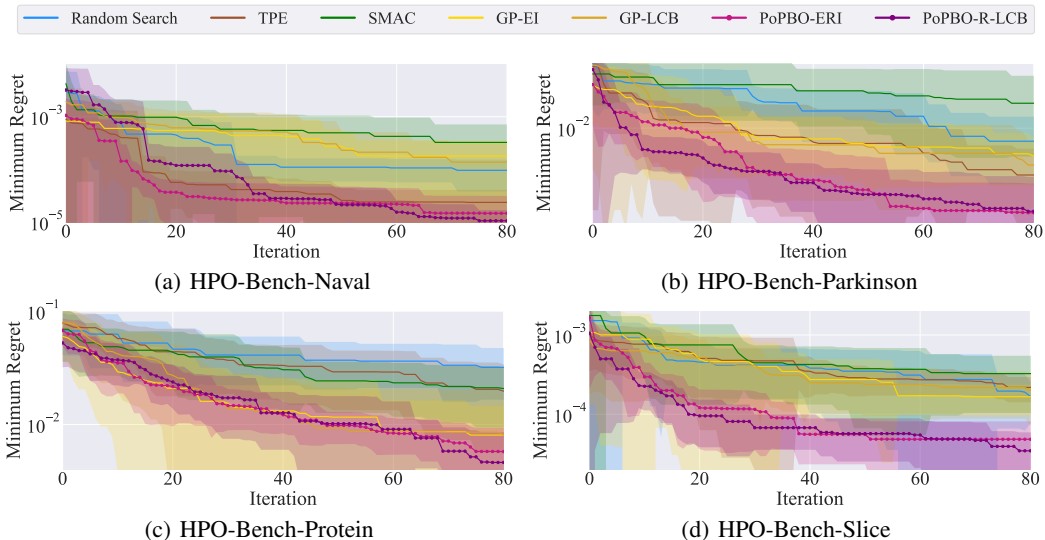

Figure 4: Minimum regret comparison with random search and various Bayesian optimization methods on tabular datasets in HPO-Bench. Y-axis indicates the residual between the optimum function value and the incumbent. We run each method ten times and plot the average performance and standard deviation of the incumbent as the line and shadow. Our PoPBO can quickly discover good samples and achieves the best performance (lowest regret).

Table 1: Top-1 mean accuracy (%) for classification on NAS-Bench-201. The first block shows the performance of non-parameter sharing algorithms and various Bayesian optimization methods. The second block shows the performance of PoPBO with ERI and R-LCB acquisition functions. [†]: Results are obtained from NAS-Bench-201. Otherwise, we independently run the method for ten times. The best mean accuracy in each column is in bold.

| Methods | CIFAR-10 | | CIFAR-100 | | ImageNet-16-120 | |
|---|---|---|---|---|---|---|
| | valid | test | valid | test | valid | test |
| REINFORCE (1992)[†] | 91.09±0.37 | 93.85±0.37 | 71.61±1.12 | 71.71±1.09 | 45.05±1.02 | 45.24±1.18 |
| REA (2019)[†] | 91.19±0.31 | 93.92±0.30 | 71.81±1.12 | 71.84±0.99 | 45.15±0.89 | 45.54±1.03 |
| Random Search (2012) | 91.00±0.38 | 93.83±0.31 | 71.29±1.29 | 71.47±1.16 | 44.83±1.11 | 45.05±1.14 |
| BOHB (2018)[†] | 90.82±0.53 | 93.61±0.52 | 70.74±1.29 | 70.85±1.28 | 44.26±1.36 | 44.42±1.49 |
| TPE (2011) | 91.06±0.38 | 93.90±0.34 | 71.29±1.29 | 71.85±1.13 | 45.04±1.23 | 45.27±1.49 |
| SMAC (2011) | 91.09±0.38 | 93.95±0.28 | 71.40±1.23 | 71.66±1.11 | 45.11±0.99 | 45.32±1.09 |
| GP (EI) (2012) | 91.40±0.18 | 94.23±0.15 | 72.67±0.83 | 72.75±0.47 | 45.83±0.45 | 46.20±0.63 |
| GP (LCB) (2012) | 91.30±0.25 | 93.98±0.22 | 72.00±0.80 | 72.05±0.76 | 45.37±0.83 | 45.60±0.94 |
| BOHAMIANN (2016) | 91.36±0.16 | 94.13±0.23 | 72.36±0.82 | 72.38±0.81 | 45.93±0.66 | 46.18±0.60 |
| PoPBO (ERI) | **91.52±0.05** | **94.35±0.03** | **73.21±0.29** | **73.25±0.18** | **46.27±0.36** | 46.54±0.19 |
| PoPBO (R-LCB) | **91.52±0.04** | 94.33±0.08 | **73.21±0.36** | 73.19±0.31 | 46.12±0.43 | **46.61±0.32** |

outperforms the state-of-the-art Bayesian optimization methods BOHB (Falkner et al., 2018) and BOHAMIANN (Springenberg et al., 2016). Additionally, we plot the performance trend of various methods on the test set of CIFAR-10, CIFAR-100, and ImageNet16-120 in Fig. 5. We observe that though our PoPBO slightly falls behind GP at an early stage, it catches up with GP at around 30-th epoch and takes the lead till the end of the search procedure. The performance trend on the validation set is also displayed in Fig. 7 in Appendix C.3.

## 5.3 EFFECTIVENESS OF THE RECTIFIED TECHNIQUE

The quantile parameter $q$ in Eq. 11 controls the trade-off between exploration and exploitation. A smaller $q$ has better exploration ability. On the one hand, it is undesirable to set $q$ as a rather small value since our method degrades to Random Search when $q \rightarrow 0$, making PoPBO suffers an

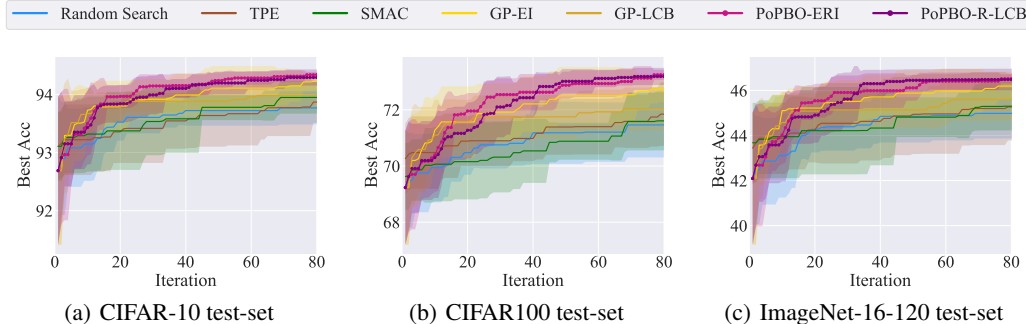

Figure 5: Performance trend of random search and Bayesian optimization methods on NAS-Bench-201. We run 10 times for each setting and plot the mean accuracy as the lines. Note that when testing Random Search, GP-BO, and our PoPBO, we adopt the same initial random seeds for all settings at each run for fairness. Hence, the lines in each plot have the same initial point (at the 0-th iteration).

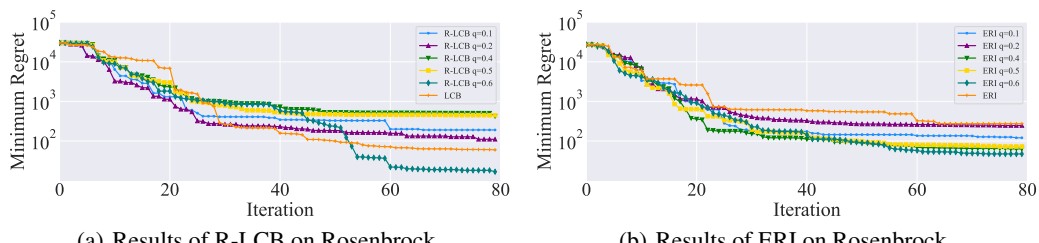

Figure 6: Ablation study on hyperparameter $q$, which controls the exploitation-exploration trade-off. We test on the Rosenbrock simulation function, a complex landscape, and show the effect of $q$ on (a) R-LCB and (b) ERI. For each setting, we run ten times and plot the average performance of the incumbent. Notice that we adopt the same initial random seeds for all settings at each run for fairness. Hence, the lines in each plot have the same initial point (at the 0-th iteration).

over-exploration issue. On the other hand, our method degrades to the vanilla acquisition functions when $q \to 1$, making PoPBO suffers an over-exploitation issue as our analysis in Sec. 4. Fig. 6 evaluates the effect of quantile parameters $q$ on both R-LCB and ERI. We independently run ten times for each setting and adopt the same initial random seeds for all settings at each run for a fair comparison. Hence, the lines in Fig. 6 have the same initial point (at the 0-th iteration). We discover that the best setting of the quantile parameter $q$ for R-LCB is $0.6$, and the best for ERI is $0.4$.

## 6 CONCLUSION

We have proposed a novel Bayesian Optimization framework, named PoPBO, for optimizing black-box functions with relative responses, being more robust to noise than absolute responses. Specifically, we introduce a relative response surface to capture the global ranking of candidates based on the Poisson process that is suitable for modeling discrete count events. We give the likelihood and posterior forms of ranking under the general assumption of a non-homogeneous Poisson process. To balance the trade-off between exploration and exploitation, we design two acquisition functions, namely Rectified Lower Confidence Bound (R-LCB) and Expected Ranking Improvement (ERI), for our ranking-based response surface. Our method enjoys a lower computational complexity of $O(N^2)$ compared to GP's $O(N^3)$ and performs competitively on both simulated and real-world benchmarks.

**Limitations and Future Work.** This work analyzes the robustness of relative response against noise and thus does not involve prior knowledge of noise. However, there exist real scenarios where the noise is too large to disrupt the ranking of observations, and we would like to leave it as our future work. Additionally, the mean of Poisson distribution is the same as variance, which has a potential over-exploitation issue as mentioned in Sec. 4. This work introduces a rectified technique to alleviate it, and we would like to explore other elegant acquisition functions in future work.

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

## A    DISCUSSION ON THE ASSUMPTIONS FOR $\hat{R}_x(S)$

We assume $\hat{R}_x(S)$ has the following properties: 1) $\hat{R}_x(\varnothing) = 0$ and 2) $\lim_{\Delta s \to 0} P(\hat{R}_x(S') + \Delta s) - \hat{R}_x(S') \geq 2) = 0, \forall S' \subset S$. The first assumption is naturally satisfied since there are no points $x' \in \varnothing$ satisfying $f(x') > f(x)$, where $f(x)$ indicates the observation of point $x$. The second assumption is naturally satisfied when the black-box function has a discrete domain of definitions since we can find a small enough $\Delta s$ that only contains one point. We further assume that it still holds in the case of continuous spaces since we have no prior information about the black-box function whose observation is noisy and thus discrete everywhere. Specifically, we introduce the following proposition.

**Proposition 1.** *Suppose a black-box function $\hat{f}(x)$ whose observation $f(x) = \hat{f}(x) + \epsilon$ has additive noise $\epsilon \sim \mathcal{N}(0,\sigma)$ obeying Gaussian distribution. Then the observation $f(x)$ is discrete everywhere.*

*Proof.* We would like to first give the definition of continuous, and then provide the definition of discrete through the negative continuous proposition.

*Definition of continuous.* If $\forall \mu > 0, \exists \delta > 0, \forall x' \in \delta(x) \triangleq \{x' : |x' - x| < \delta\}$ satisfying $|f(x') - f(x)| < \mu$, we say $f(x)$ is continuous at point $x$.

*Definition of discrete (negative continuous proposition).* If $\exists \mu > 0, \forall \delta > 0, \exists x' \in \delta(x)$ satisfying $|f(x') - f(x)| \geq \mu$, we say $f(x)$ is discrete (not continuous) at point $x$.

Consider proposition 1, $\forall \delta, \forall x' : |x' - x| < \delta, |f(x') - f(x)| = |\hat{f}(x') - \hat{f}(x) + \epsilon' - \epsilon|$, where $\epsilon', \epsilon$ indicates the observation noise at $x', x$. $\Delta \epsilon = (\epsilon' - \epsilon) \sim \mathcal{N}(0, 2\sigma)$ also obeys Gaussian distribution since $\epsilon'$ and $\epsilon$ are independent stochastic variables obeying Gaussian distribution. Hence, $P(\exists x' \in \delta(x), |f(x') - f(x)| \geq \mu) = 1 - P(\forall x' \in \delta(x), |f(x') - f(x)| < \mu) = 1 - \prod_{x' \in \delta(x)} P(\Delta \epsilon \in (-\mu - \hat{f}(x') + \hat{f}(x), \mu + \hat{f}(x') - \hat{f}(x))) \approx 1$, since $f(x)$ is continuous in $\delta(x)$. Therefore, we can say that $f(x)$ is almost surely discrete anywhere.

In our implementation, $\hat{R}_x(S)$ depends on a discrete set $S \cap \hat{S}$. We can find a small enough $\Delta s$ satisfying $\Delta s \cap \hat{S} = \varnothing$. Therefore, the property 2 (sparse assumption) is naturally satisfied in our implementation.

## B    ALGORITHM DETAILS

As the number of observations $N$ increases, the right truncated Poisson distribution gradually approaches to the normal one(Yigiter & Inal, 2006), which has a smaller computational cost. Hence, we use normal Poisson process to model the response surface when $N \geq 12$.

## C    SUPPLEMENTARY OF EXPERIMENTS

### C.1    EXPERIMENTAL SETTINGS ON ROBUSTNESS ANALYSIS

We compare the sensitivity to additive Gaussian noise between GP (value-based) response surface and PoPBO (ranking-based) response surface on Forrester function. To simulate the performance of GP, we first utilize Gaussian process to fit a certain number of (15 in this paper) observed values and plot the ranking of e.g. 100 points according to the values predicted by GP as Fig. 1(a). Meanwhile, our method utilizes Poisson process to directly capture the ranking response surface based on the same 15 observations and predict the ranking of 100 points as shown in Fig. 1(b). We observe that our response surface is more robust to noise and can better capture the global ranking.

### C.2    DETAILED SETTINGS OF BASELINE METHODS

In this section, we provide the specific details of each baseline mentioned in the paper:

---

[1]Specifically, a sorting function can serve as the $Rank(\cdot)$.

---

**Algorithm 1:** PoPBO: Bayesian Optimization with Poisson Process

**Inputs :**

      1) A function $Rank(\{\cdot\})$ [3] to rank samples based on a black-box function $f(\cdot)$;

      2) A feasible domain (search space) $X$;

      3) An acquisition function $\alpha$ (R-LCB as Eq. 11 or ERI as Eq. 12);

      4) The number of initial points $N$;

      5) The number of total training iterations $T$

1   Random sample $N$ initial points $\hat{S} \coloneqq \{x_j\}_{j=1}^{N}$ from $X$;

2   Initialize the parameters $\theta$ of $\lambda_\xi(x)$;

3   **for** $t \coloneqq from\ 1\ to\ T$ **do**

4      Get their rankings $\hat{R} \coloneqq Rank(\hat{S})$;

5      Learn $\theta$ based on points $\hat{S}$ and $\hat{R}$ by minimizing Eq. 7 with gradient as Eq. 8;

6      Get the next query $x^*$ by minimizing acquisition function $\alpha$;

7      Update the set of points $\hat{S} \coloneqq \hat{S} \cup \{x^*\}$

8   **end**

9   Get the best query $x_{opt}$ in history based on $Rank(\hat{S})$;

10   **Output:** The best point $x_{opt}$ in history.

---

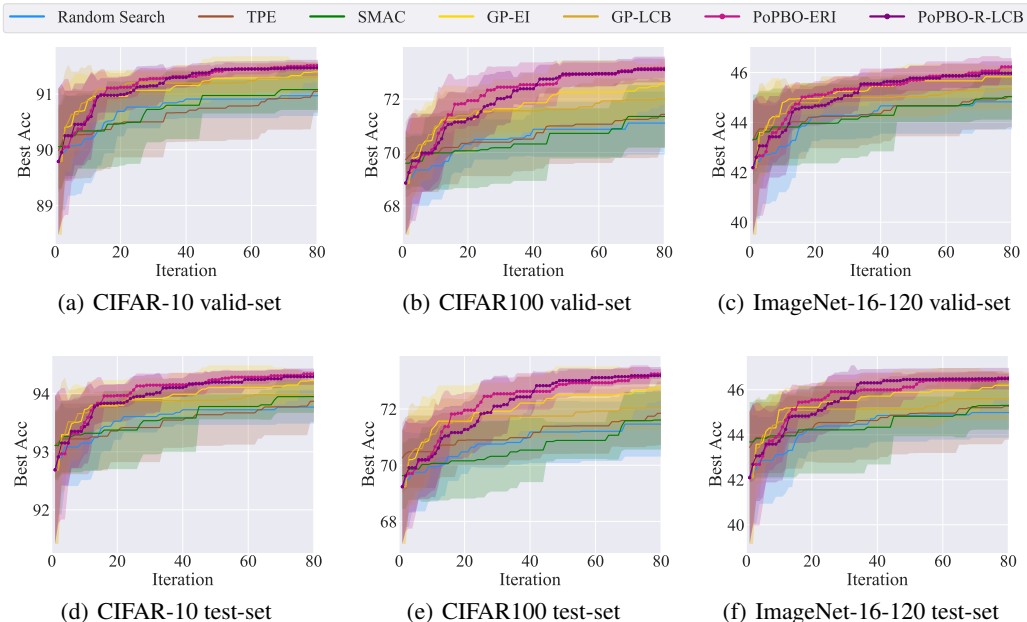

(a) CIFAR-10 valid-set      (b) CIFAR100 valid-set      (c) ImageNet-16-120 valid-set

(d) CIFAR-10 test-set      (e) CIFAR100 test-set      (f) ImageNet-16-120 test-set

Figure 7: Performance trend of random search and Bayesian optimization methods on NAS-Bench-201. We run 10 times for each setting and plot the mean accuracy as the lines. Note that when testing Random Search, GP-BO, and our PoPBO, we adopt the same initial random seeds for all settings at each run for fairness. Hence, the lines in each plot have the same initial point (at the 0-th iteration).

**Random Search (RS).** Following the description in Bergstra & Bengio (2012), we sample candidates uniformly at random.

**BO with Gaussian Process (GP).** We follow the settings that described by Snoek et al. (2012) and use the implementation of our own. We use expected improvement (EI) and lower confidence bound (LCB) as acquisition functions and adopt LBFGS to optimize them. When the search space is completely discrete like Dong & Yang (2020), we use random sampling to find the next query, which gives the maximizer of the acquisition function among $N = 1000$ random samples. For kernel function, we use ARD Matérn 5/2 kernel for GP. During the training process, we adopt slice sampling,

| Methods | Naval | Parkinson | Protein | Slice |
|---|---|---|---|---|
| Random Search (2012) | $9.54 \times 10^{-5} \pm 5.76 \times 10^{-5}$ | $7.99 \times 10^{-3} \pm 4.18 \times 10^{-3}$ | $3.04 \times 10^{-2} \pm 1.51 \times 10^{-2}$ | $1.48 \times 10^{-4} \pm 7.45 \times 10^{-5}$ |
| TPE (2011) | $1.67 \times 10^{-5} \pm 1.64 \times 10^{-5}$ | $4.49 \times 10^{-3} \pm 2.55 \times 10^{-3}$ | $1.86 \times 10^{-2} \pm 1.37 \times 10^{-2}$ | $1.80 \times 10^{-4} \pm 7.10 \times 10^{-5}$ |
| SMAC (2011) | $2.94 \times 10^{-4} \pm 3.96 \times 10^{-4}$ | $1.48 \times 10^{-2} \pm 9.38 \times 10^{-3}$ | $2.05 \times 10^{-2} \pm 1.14 \times 10^{-2}$ | $3.16 \times 10^{-4} \pm 2.25 \times 10^{-4}$ |
| GP (EI) (2012) | $1.79 \times 10^{-4} \pm 1.77 \times 10^{-4}$ | $6.20 \times 10^{-3} \pm 2.44 \times 10^{-3}$ | $8.06 \times 10^{-3} \pm 6.53 \times 10^{-2}$ | $1.66 \times 10^{-4} \pm 7.00 \times 10^{-5}$ |
| GP (LCB) (2012) | $1.38 \times 10^{-4} \pm 1.77 \times 10^{-4}$ | $5.31 \times 10^{-3} \pm 2.79 \times 10^{-3}$ | $8.08 \times 10^{-3} \pm 1.02 \times 10^{-2}$ | $1.95 \times 10^{-4} \pm 1.30 \times 10^{-4}$ |
| PoPBO (ERI) | $1.38 \times 10^{-5} \pm 1.59 \times 10^{-5}$ | $\mathbf{1.92 \times 10^{-3} \pm 1.51 \times 10^{-3}}$ | $5.77 \times 10^{-3} \pm 3.55 \times 10^{-3}$ | $4.83 \times 10^{-5} \pm 3.29 \times 10^{-5}$ |
| PoPBO (R-LCB) | $\mathbf{1.07 \times 10^{-5} \pm 6.26 \times 10^{-6}}$ | $2.41 \times 10^{-3} \pm 1.93 \times 10^{-3}$ | $\mathbf{4.62 \times 10^{-3} \pm 3.91 \times 10^{-3}}$ | $\mathbf{3.14 \times 10^{-5} \pm 1.74 \times 10^{-5}}$ |

Table 2: Regret of the configuration discovered by various methods on the four datasets of HPO-Bench. We run each method for ten times and report the mean and standard deviation. The best performance (lowest mean value and standard deviation) is in bold.

an efficient Markov chain Monte Carlo (MCMC) method, to fit the hyperparameters of GP, which we find to work more robustly for GP.

**Tree Parzen Estimator (TPE).** Bergstra et al. (2011) adopt kernel density estimators to model the probability of points with bad and good performance respectively. Then TPE give the next query by optimizing the ratio between the two estimated likelihood, which is proved to be equivalent to optimizing EI. We use the default settings provided in hyperopt package (`https://github.com/hyperopt/hyperopt`).

**SMAC.** Hutter et al. (2011) adopt random forest to model the response surface of the black-box function. We use the default settings given by scikit-optimize package (`https://github.com/scikit-optimize/scikit-optimize`).

**BOHAMIANN.** Unlike Snoek et al. (2012), BOHAMIANN adopts bayesian neural network to build the response surface, whose weights are sampled via a stochastic gradient Hamiltonian Monte-Carlo (SGHMC) method. We use the default settings provided in pybnn package (`https://github.com/automl/pybnn`) and use EI as the acquisition function.

**PPBO.** This is an effective preferential BO method based on pairwise comparisons, attempting to learn user preferences in high-dimensional spaces. We use the default settings provided by PPBO package (`https://github.com/AaltoPML/PPBO`).

**HEBO.** Heteroscedastic Evolutionary Bayesian Optimisation that won the NeurIPS 2020 black-box optimisation competition. We use the default strategy and its default parameters provided in HEBO package (`https://github.com/huawei-noah/HEBO`). For a fair comparison, we use a uniform sampling strategy instead of a sobol one during initialization and candidates generation.

## C.3    DETAILED RESULTS ON HPO-BENCH AND NAS-BENCH-201

Table 2 reports the numerical performance of PoPBO and other methods on the four datasets on HPO-Bench. Fig. 7 displays the performance trend of PoPBO and other methods on the validation set and test set under the search space of NAS-Bench-201.

## C.4    IS THE GAIN IN PERFORMANCE OF PoPBO DUE TO THE COMPLEX REPRESENTATION OF MLP IN THE SURROGATE MODEL?

We utilize the same MLP architecture and training settings as PoPBO's to fit the Gaussian likelihood of each candidate. We denote such a setting as GP-MLP. We plot the regret of GP-MLP, GP, and our PoPBO on Hartmann and Rosenbrock in Fig. 8. We observe that GP-MLP performs much worse than GP and ours, showing that the complex representation of the MLP in the surrogate model is not the main reason for the gain in performance.

## C.5    ABLATION STUDY ABOUT THE WORST TOLERANT RANKING $K_m$ IN ERI

Fig. 9 compares the performance of ERI under various settings for the worst tolerant ranking $K_m$. We observe that PoPBO-ERI is not very sensitive to $K_m$. Specifically, for Branin, ERI with larger

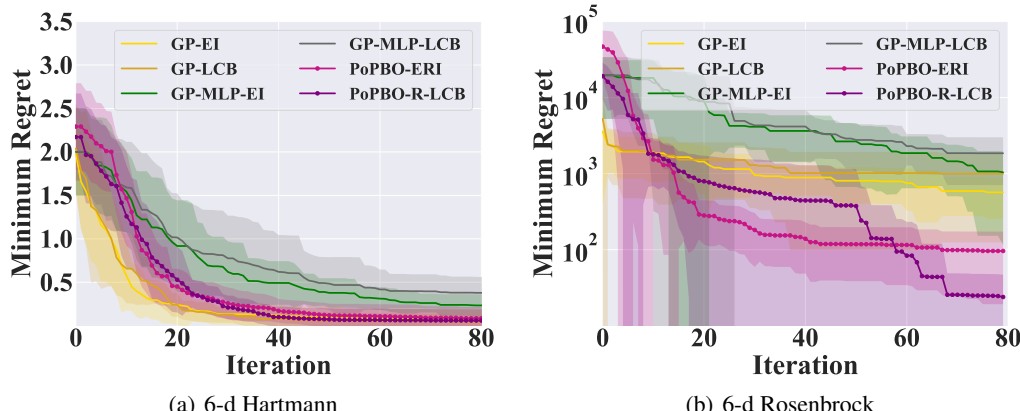

Figure 8: Performance trend of GP, GP-MLP, and PoPBO on Hartmann and Rosenbrock simulated benchmarks.

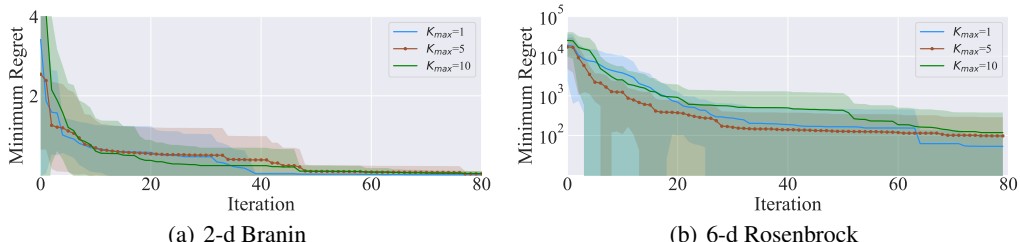

Figure 9: Ablation study of ERI on the two simulation functions. $K_{max}$ is the worst tolerable ranking, and a higher $K_{max}$ leads to a higher rate of exploration.

$K_m$ converges faster but ERI with various $K_m$ have similar ultimate performance after 80 iterations. For Rosenbrock that has larger search space and more complex landscape, better exploitation ability is more important than exploration, and thus ERI with lower $K_m$ achieves better performance.

## C.6 More iterations on 6-d Rosenbrock for GP and PoPBO

Since 6-d Rosenbrock has a large search space and is hard to converge, we run GP and PoPBO for more iterations (200 queries) and plot the regret in Fig. 10. We observe that PoPBO consistently outperforms GP after 50 epochs. Moreover, both PoPBO-ERI and PoPBO-R-LCB have lower variance than GP-EI and GP-LCB.

## C.7 Robustness to various noise level

In our settings, the observations in the simulation function are noiseless, while the observations in the real-world benchmark (HPO-Bench and NAS-Bench-201) are noisy. For HPO-Bench, the

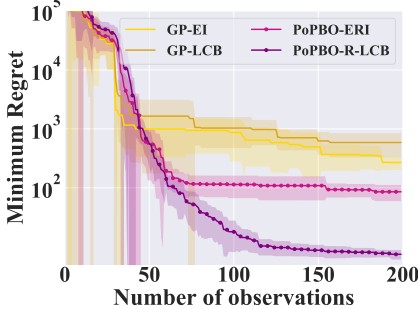

Figure 10: Performance trend of PoPBO and GP by running 200 iterations on 6-d Rosenbrock. For each setting, we conduct replicated experiments for six times with various random seeds.

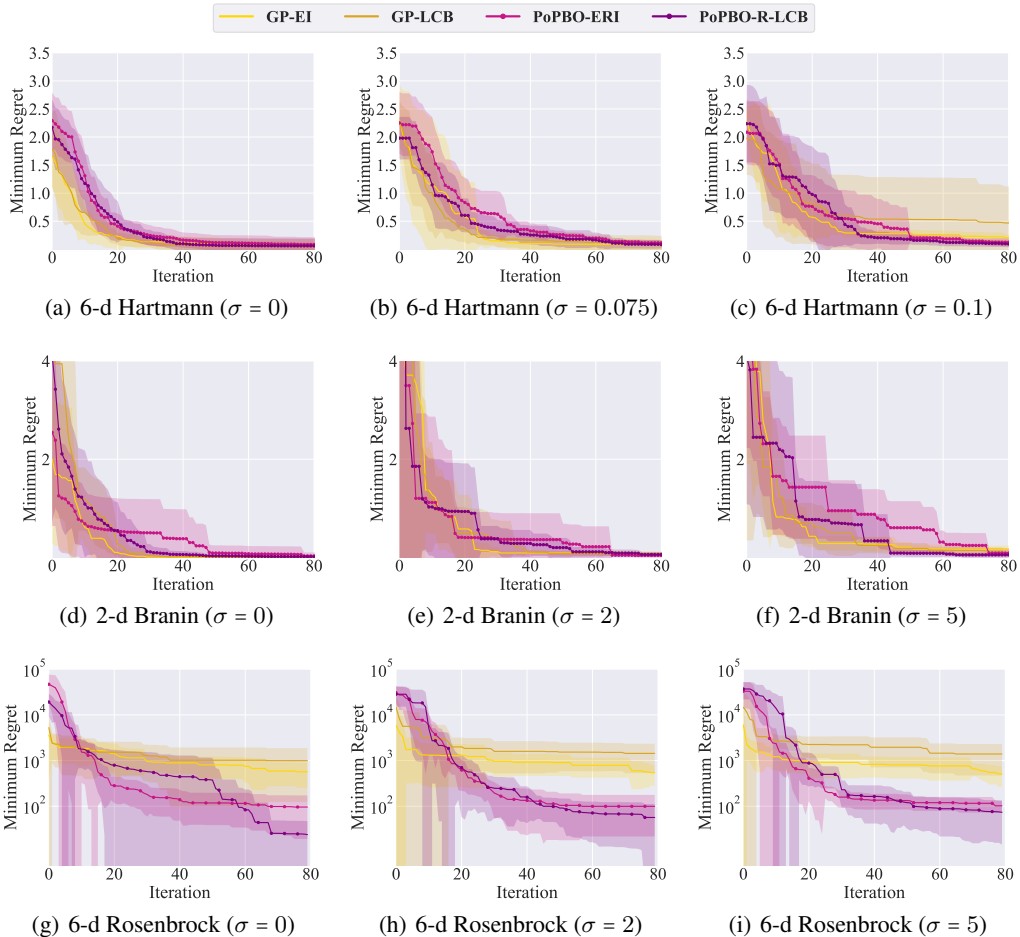

Figure 11: Comparison of the robustness to noise between GP-BO and PoPBO. We add Gaussian noises with zero mean and various standard deviation $\sigma$ to the objective function. Notice that Branin and Rosenbrock have much larger range of value than Hartman, so we add noises with larger variance to them.

performance of each configuration (hyperparameters of FCNet) is evaluated 4 times under different random seeds. In the experiment settings of HPO-Bench, the average performance is used as the observation, which is naturally noisy. NAS-Bench-201 attempts to search for a neural architecture that performs best after 200 training epochs. However, following the experiment settings of NAS-Bench-201, only the validation accuracy after 12 training epochs of each architecture can be queried, making the observations noisy.

To verify the robustness to noise of PoPBO on simulated benchmarks, we add Gaussian noises with zero mean and various standard deviation ($\sigma$) to Hartmann, Branin, and Rosenbrock simulation functions and run GP-BO and PoPBO separately. Trends of average regret among six parallel tests are plotted in Fig. 11, showing that GP performs worse with the increment of noise level. In contrast, PoPBO performs much more stable. Moreover, PoPBO outperforms GP when the objective function has large noise ($\sigma = 0.1$ for Hartmann, $\sigma = 5$ for Branin and Rosenbrock). The results demonstrate the robustness of PoPBO to noise.

## D    NOTATIONS

1  $X$ is the whole feasible domain (search space). If $X$ is continuous, $|X|$ is the volume of $X$. While $X$ is discrete, $|X|$ is the cardinality of it.

2  $S_x = \{y | y \in S, f(y) < f(x)\}$ is the set of better points than $x$ in $S \subset X$, where $S$ can be any continuous domain or discrete set.

3 $\hat{S}$ is a discrete set containing both initial samples for BO and the history queries.

4 Given a specific $S$, $\hat{R}_x(S)$ is a random variable denoting possible ranking of $x$ over a discrete set $\hat{S}_x \cap \hat{S}$. Hence, $\hat{R}_x(S)$ depends on $\hat{S}$, we utilize a hat symbol ˆ to simplify the notation.

## E  BROADER IMPACT AND LIMITATIONS

This paper addresses the problem of Bayesian Optimization (BO) to enable efficient and effective black-box optimization. It has broad applications in perception tasks, especially computer vision, robotic control and biology. This, on the one hand, would facilitate our daily life, and on the other hand, we shall be careful about their abuse which may break one's privacy. In this sense, privacy-protection BO is also needed for development, and our techniques can also be of specific help for its generality.

