# OpenReview forum: "Poisson Process for Bayesian Optimization"
_ICLR.cc/2023/Conference — Submitted to ICLR 2023_

### Official Review · Reviewer_55na · 2022-10-21

**Confidence:** 4
**Correctness:** 2
**Technical Novelty And Significance:** 2
**Empirical Novelty And Significance:** 2
**Recommendation:** 5

**Clarity, Quality, Novelty And Reproducibility:**

The paper writing is also easy to understand in general (except the mathematical formulas which are confusing) - please refer to my comments above.

I have various concerns regarding the quality of the paper - please refer to my comments above.

I think the main idea of the paper is interesting and new - please refer to my comments above.

There is no code uploaded as part of the submission.

**Strength And Weaknesses:**

Strengths:
+ Overall, I think the idea of modelling the relative response and use this model to guide the BO process is interesting and new.
+ The paper writing is also easy to understand in general (except the mathematical formulas).
+ The proposed method has good performance on some problems.

Weaknesses: I still have various concerns regarding the paper, especially, its rigorousness. Please find in the below my concerns:
+ The sentences "Considering the physical meaning of S_x, we can estimate the superiority of x for f(⋅) against the points in set S by measuring S_x. Specifically, considering two points x1, x2, S_x1 has a larger measure value, representing that there are more points in S better than x1, so x1 is worse than x2." in Section 3.1 are ambiguous and are not clear to understand. What do we mean by "measuring S_x", S_x is a set, so what property we want to measure? Here, I think the paper means to say we need to come up with a metric/value to quantify S_x?

+ The important notations in the paper are unclear and confusing, which make it hard to understand the maths behind the proposed method. For example, I don't understand Eq. (2), in particular, why the notation \hat{R}_x(S) does not depend on \hat{S}. Does this mean for any set of sampled candidates \hat{S}, the value is \hat{R}_x(S) is the same? Later, in Section 3.2, whenever the term \hat{R}_x(S) is mentioned, there is no mention about \hat{S} anymore. This really confuses me. Also, if I read the notation correctly, is Q a set of sets? Also, what is a set of sampled candidates (the notation \hat{S} in the paper)?
+ I also have more concerns regarding the measure \hat{R}_x(S). For continuous domain X, will \hat{R}_x(S) be infinite? As the cardinality of all the sets will be infinite. In this case, then I don't understand how the proposed method will work.
+ In Figure 1, the paper illustrates the relative response surface with a Gaussian Process, but later, a component of the Poisson process is approximated by MLP, so I don't know if the gain in performance is due to the relative response (ranking) idea or due to the complex representation of the MLP in the surrogate model.
+ In the R-LCB acquisition function, in Eq. (11), what is \epsilon_x? how to compute it?
+ In the ERI acquisition function, in Eq. (12), K_m is a hyperparameter. How is the performance of BO sensitive to this value? More discussion is needed.
+ For the 6-d Rosenbrock function, I expect more iterations (e.g., 20-30d) to be conducted as 80 iterations are a bit small compared to the dimension and complexity of this function.
+ Most importantly, in the experiments, are the observations of the objective functions (Branin, Hartman, Rosenbrock) noisy? And what will be the noise in the real-world benchmark problems? I can't find any information in the main paper. This is critical as robustness to noise is the main argument of the paper regarding the advantage of the proposed method over the standard BO methods.

Minor comments:
+ Some definitions about "relative evaluation" and "absolute response" should be explained early in the paper. In the introduction section, when these terms are first mentioned, I had some difficulties in understanding what they mean.
+ I think the relative evaluation is still sensitive to noise, it's just not as severe as absolute response. So, it's better to say it is harder to be disrupted to noise (not hard to be disrupted by noise as in many places in the paper).
+ I think the statement about "absolute response surface has poor transferability" is too overstated. It has been shown that warm-starting (reuse the data from other tasks) is very useful in BO and HPO, therefore, this shows that it is possible to transfer knowledge from other tasks to a new task in the BO or HPO process.
+ The contribution "Efficient BO Framework" seems to be too generic. Here, the paper means computational efficient?
+ First sentence of Section 3.1: block-box --> black-box


**Summary Of The Paper:**

The paper proposes to use relative function evaluation, in particular, ranking, to guide the Bayesian optimization (BO) process. The argument is that relative response is more robust to noise than absolute response, thus, if we model the objective function by relative response, the performance of the BO process might be better. The paper proposes to use Poisson Process to model this relative response, and then propose two new acquisition functions for their proposed method that are based on the LCB and EI acquisition functions. Finally, the paper evaluates the proposed method on some synthetic and real-world problems.

**Summary Of The Review:**

Even though I like the main idea of the paper, but I still have various concerns regarding the rigorousness and soundness of the proposed method. These concerns include both the theoretical and empirical aspects of the paper. Please refer to my comments above.

---

> ### Author Response · Authors · 2022-11-14
> **Reply to Reviewer 55na [part 1]**
>
> ### 1. Questions on the PoPBO
>
> **Q1: Does "measuring $S_x$" mean to come up with a metric/value to quantify $S_x$?**
>
> A1: We define $\hat{R}_x(S)$ as a metric to quantify $S_x$ by counting the number of points in $\hat{S}\cap S$, shown as Eq. 2.
>
> **Q2: 1) Does the notation $\hat{R}_x(S)$ depend on $\hat{S}$? 2) Is $Q$ a set of sets? 3) What is a set of sampled candidates (the notation $\hat{S}$ in the paper)?**
>
> A2:
> 1)$\hat{R}_x(S)$ depends on $\hat{S}$ (please see Eq. 2) and we omit it for conciseness.
> 2)$Q$ is a set of sets $\{S_x|\forall S\subset X, \forall x\in X\}$.
> 3)$\hat{S}$ contains the initial candidates and all history queries. Hence, the number of $\hat{S}$ gradually increases during the BO process.
>
> **Q3: For continuous domain $X$, will $\hat{R}_x(S)$ be infinite?**
>
> A3: $\hat{R}_x(S)= |S\cap\hat{S}|$ (Eq. 2) is finite since $hat{S}$ is discrete and finite.
>
> **Q4: Is the gain in performance due to the relative response (ranking) idea or due to the complex representation of the MLP in the surrogate model?**
>
> A4: We utilize the same MLP architecture and training settings as PoPBO's to fit the Gaussian likelihood of each candidate. We plot the regret of GP-MLP, GP, and our PoPBO in Fig. 8 (supplementary) and analyze the results in Sec. C.4 We observe that GP-MLP performs much worse than GP and ours, showing that the complex representation of the MLP in the surrogate model is not the main reason for the gain in performance.
>
> **Q5: In the R-LCB acquisition function, in Eq. (11), what is $\epsilon_x$? how to compute it?**
>
> A5: $\epsilon_x$ is a uniform random variable defined in Eq. 11 ($\epsilon_x \sim \mathcal{U}[0,1]$) and is obtained by randomly sampling. It serves for reparameterization, which is a sampling method enabling $\alpha_{R-LCB}(x)$ differentiable with reference to $x$. The Reparameterization technique is widely-used in reinforcement learning [1] and VAE [2].
>
> [1] Haarnoja T, et al. Off-policy maximum entropy deep reinforcement learning with a stochastic actor. ICML 2018.
> [2] Diederik P. Kingma and Max Welling. Auto-Encoding Variational Bayes. ICLR 2014.
>
> **Q6: $K_m$ is a hyperparameter in ERI. Is the performance of PoPBO-ERI sensitive to $K_m$?**
>
> A6: We conduct an ablation study on $K_m$ on Branin and Rosenbrock simulated benchmarks, report the results in Fig. 9 (supplementary) and analyze in Sec. C.5. We observe that PoPBO-ERI is not very sensitive to $K_m$. Specifically, for Branin, ERI with larger $K_m$ converges faster but ERI with various $K_m$ have similar ultimate performance after 80 iterations. For Rosenbrock which has a larger search space and more complex landscape, better exploitation ability is more important than exploration, and thus ERI with lower $K_m$ achieves better performance.
>
>
> **Q7: For the 6-d Rosenbrock function, I expect more iterations (e.g., 20-30d) to be conducted as 80 iterations are a bit small compared to the dimension and complexity of this function.**
>
> A7: we run GP and PoPBO for 200 iterations and plot the regret in Fig. 10 (supplementary) and analyze in Sec. C.6. We observe that PoPBO consistently outperforms GP after 50 epochs. Moreover, both PoPBO-ERI and PoPBO-R-LCB have lower variance than GP-EI and GP-LCB.
>
> **Q8: What is the noise in the simulation experiments and the real-world benchmark problems? How about the experiments on the robustness of PoPBO to noise?**
>
> A8: The observations in the simulation function are noiseless, while the observations in the real-world benchmark (HPO-Bench and NAS-Bench-201) are noisy. For HPO-Bench, the performance of each configuration (hyperparameters of FCNet) is evaluated 4 times under different random seeds. In the experiment settings of HPO-Bench, the average performance is used as the observation, which is naturally noisy. NAS-Bench-201 attempts to search for a neural architecture that performs best after 200 training epochs. However, following the experiment settings of NAS-Bench-201, only the validation accuracy after 12 training epochs of each architecture can be queried, making the observations noisy.
>
> Moreover, we conduct experiments on noisy simulation functions with various noise levels, whose results are shown in Fig. 11 (supplementary) and analyzed in Sec. C.7. Specifically, we observe that PoPBO outperforms GP when the objective function has large noise ($\sigma=0.1$), demonstrating the robustness of PoPBO to noise.

---

> > ### Author Response · Authors · 2022-11-14
> > **Reply to Reviewer 55na [part 2]**
> >
> > ### 2. Comments on the Writing
> >
> > **Q1: Definitions of "relative evaluation" and "absolute response" should be explained early.**
> >
> > A1: Thanks for your suggestion. We add footnotes to define them.
> >
> > **Q2: It's better to say it is harder to be disrupted by noise (not hard to be disrupted by noise as in many places in the paper).**
> >
> > A2: Thanks for your suggestion. We have refined it in the new version.
> >
> > **Q3: The statement about "absolute response surface has poor transferability" is too overstated.**
> >
> > A3: Thanks for your kind suggestion, and we refine it in the new version. Nevertheless, we have learned that many BO methods for transfer learning try to involve rankings metrics to conduct the transfer process, such as [1,2,3]. Unlike the prior works that build absolute response surfaces, our method attempts to directly capture the relative response, which we believe can have more potential transferability. We would like to explore it in our future work.
> >
> > [1] David Salinas, Huibin Shen, and Valerio Perrone. A quantile-based approach for hyperparameter transfer learning. In ICML, 2020.
> > [2] Feurer M, Letham B, and Bakshy E. Scalable meta-learning for bayesian optimization using ranking-weighted gaussian process ensembles. In ICML AutoML Workshop. 2018.
> > [3] Bardenet R, et al. Collaborative hyperparameter tuning. In ICML, 2013.
> >
> > **Q4: The contribution "Efficient BO Framework" seems to be too generic. Here, the paper means computational-efficient?**
> >
> > A4: Thanks for your advice. This paper means computational-efficient. We make it clear in the new version.
> >
> > **Q5: First sentence of Section 3.1: block-box --> black-box**
> >
> > A5: Thanks for pointing out the typo, we have refined it in this version.

---

> ### Comment · Reviewer_55na · 2022-12-01
> **Response to the authors' rebuttal**
>
> Dear authors,
>
> Thank you for your response. I've gone through your response and also have looked at the revised version of the paper. However, my concerns are still there. In particular, there is still a lot of confusion regarding the mathematical notations and unclear expression of the proposed method. These are very important to ensure the correctness of the proposed method. Therefore, I decided to still keep my original score of the paper, which is 6.
>
> Warm regards,

---

### Official Review · Reviewer_dFd1 · 2022-10-21

**Confidence:** 3
**Correctness:** 2
**Technical Novelty And Significance:** 4
**Empirical Novelty And Significance:** 2
**Recommendation:** 5

**Clarity, Quality, Novelty And Reproducibility:**

As mentioned in 'Weaknesses', I think there is room to improve for a better presentation of the idea. I guess the suggested 'point process interpretation' may help to resolve to improve the clarity.
Among others, I value the novelty of the Poisson process surrogate model.
The Poisson process has been studied extensively and there are many extensions of it, this paper can stimulate the development of various surrogate models utilizing the theory of Poisson processes.
Despite my concern about the explanation of the surrogate model, all implementation detail and algorithms seem to be provided with enough detail to try this PoPBO.
Along with detailed information on the implementation, the empirical analysis is reasonable enough to show the superiority of the method accompanied by many ablation studies.


**Strength And Weaknesses:**

### Strengths
- The paper provides a novel surrogate model for ranking which has many advantages over conventional surrogate models for values.
- Two acquisition functions utilizing a ranking-based surrogate model are proposed.
- Many aspects of the proposed surrogate model and acquisition functions are empirically covered by ablation studies.
- Empirically, rank-based BO using the Poisson process surrogate model is shown to outperform existing methods consistently.
- PoPBO avoids the cubic complexity coming from Gaussian processes, which enables a significant reduction in runtime.


### Weaknesses
- The explanation of the Poisson process surrogate model in 3.1, 3.2, and Appendix A is a bit unclear. (However, in the 'point process interpretation' below, I suggest a different perspective that may allow dismissing the below issues. The authors may want to check that first.)
    - In Eq.(2), does the definition of $\hat{R}_x$ depend on the choice of $\hat{S}$? This does not seem a natural way to define Poisson processes.
    - In the proof of Prop.1, what does the first line mean? Even without $|x'-x| < \delta$, it holds that $f(x')-f(x) = \hat{f}(x')-\hat{f}(x)+\epsilon-\epsilon'$ for all $x$ and $x'$. Why is $\delta$ necessary?
    - It argues that since $P(|f(x') - f(x)| = 0) = 0$, $f(x)$ is discrete. But for $f(x)=x$ and $\sigma=0$, we can say $P(|f(x') - f(x)| = 0) = 0$ as long as $x \neq x'$, but $f$ is not discrete.
    - It seems that the inverse of the $\epsilon$-$\delta$ method was tried in Prop.1. What was tried to prove is that $f$ is discontinuous everywhere? If not, what is the rigorous definition of 'discrete'? For example, there should be something like 'For $a \in A \subset R^N$, $a$ is a discrete point of $A$ if there exists an open neighborhood of $a$ which does not intersect with $A$'.
    - Also, it is not clear how to conclude the property 2) is derived from Prop.1. Can you elaborate on this?
- Confusing notations hinder understanding of the idea.
    - In the 4-th line of 3.2, $\hat{R}_x (S + \Delta s | x) - N(S|x)$ seems to mean $\hat{R}_x (S + \Delta s) - \hat{R}_x (S)$.
      In Appendix A, $N(S|x)$ is used.
      For the reasons I elaborate below, it seems that using the notation for the count $N(S|x)$ is more natural.
    - In many lines, $| \cdot |$ is used for both finite sets and infinite sets. I guess $|\hat{S}|$ is the number of elements and $|X|$ is the volume of  $X$.
    - In Eq.(2), I guess it meant $| S_x \cap \hat{S} |$ without curly brackets.
    - In Eq.(4), since for a given set $S$, $\hat{R}_x(S)$ is a Poisson random variable, it should be either
$$
\hat{R}_x(S) \sim Poisson(\int_S \lambda(s, x) ds) \quad \text{OR} \quad \hat{R}_x \sim \mathcal{P}_n(\lambda(s,x))
$$


### Other questions
- **How good is the uncertainty of the Poisson process surrogate model?**
    - It seems that the trick 'rectification' is crucial in performance. I don't see this as a critical issue when claiming empirical superiority and robustness since a single choice for $q=0.6$ gives good performance on various problems.
However, it may imply that the Poisson process surrogate model may not be good as expected.
    - Even though the property of the Poisson random variable is pointed out as a possible reason in the paper, I am curious whether it is because of the way how the Poisson process surrogate model is trained.
It seems that the way the Poisson process surrogate model is trained is different from how GP, BNN, Deep ensemble, etc are trained.
GP and BNN are fitted in Bayesian way using posteriors in their prediction. Also, Deep ensemble is claimed to mimic Bayesian inference. However, the Poisson process surrogate model is a single density model fitted with MLE, so poor uncertainty may come from this training procedure.
- **Point process interpretation** In the below comments, I share my own interpretation to have a better picture of the method. Mostly I am curious about authors' opinions on theses.
    - On the surface, PP just replaces the absolute response in GP with the ranking response.
It seems that from the point process interpretation rather than ranking interpretation, the explanation of the Poisson process surrogate model can be improved.
    - At each round of PoPBO, the data used to fit the PP surrogate model is the collection of evaluations chosen by PoPBO.
It seems that the PP surrogate model this generative process.
I guess that the probability of a set $\\{x_1, \cdots, x_N \\}$ from PP model in PoPBO is
$$
\sum_{\pi \in S_N} p(x_{\pi^{-1}(1)}) \cdot p(x_{\pi^{-1}(2)} | x_{\pi^{-1}(1)}) \cdots p(x_{\pi^{-1}(N)} | x_{\pi^{-1}(N-1)}, \cdots, x_{\pi^{-1}(1)})
$$
where $p(x_{\pi^{-1}(n)} | x_{\pi^{-1}(n-1)}, \cdots, x_{\pi^{-1}(1)})$ is the probability that $x_{\pi^{-1}(n)}$ is selected as the next query given previous $n-1$ evaluations $\\{ x_{\pi^{-1}(n-1)}, \cdots, x_{\pi^{-1}(1)} \\}$ at $n$-th round of PoPBO.
Since PP does not consider a sequence but a set ignoring orders, I guess the probability will be the sum of all possible sequences but I believe that it is likely that the sequence appearing in the BO run has the highest density.
    - The point process interpretation allows a more natural definition of $\hat{R}_x$ in Eq.(2).
In Eq.(2), $\hat{R}_x(S)$ which should be a Poisson random variable by Eq.(4), depends on $\hat{S}$, the sampled candidates.
Then from this definition, it seems that the Poisson process surrogate model also depends on the choice of $\hat{S}$.
However, in the point process interpretation, $\hat{R}_x(S)$ is defined as the number of random points $S_x$ where the random points are a set of queries generated by PoPBO.
    - I guess that when $N$ evaluations are given, then the intensity $\lambda(s,x)$ is fitted so that $P(\hat{R}_x(S)=k)$ is the probability that $S_x$ contains $k$ points in the queries suggested by PopBO under the response surface model based on given evaluations, i.e. $\hat{R}_x(S)$ is the number of points in $S$ whose evaluation is better than $f(x)$, which would be suggested by PoPBO when PoPBO has its model on the response surface based on $N$ evaluations.
    - Since what the Poisson process surrogate model does is not explained in detail, I tried to reach some reasonable interpretation as above. With the above interpretation, since the Poisson process surrogate model uses $N$ evaluations to have its own picture on the response surface, I guess the current training procedure using MLE on a single probabilistic model could be a weakness of the approach, especially in uncertainty quantification.
    - With this new definition of $\hat{R}_x$ without $\hat{S}$, it seems that the truncation in Eq.(5) is not necessary anymore, which is also aligned with what is done in the implementation.
    - Moreover, the point process interpretation can remove the necessity of the justification that $\hat{R}_x$ is the Poisson process in Appendix A, which is a bit confusing.
    - With this description of what PP models, it can be interpreted that PP in PoPBO is kind of a generative model in contrast to the discriminative model of GP.
By modeling how input points are generated, the generative process may model how the next queries are chosen.

**Summary Of The Paper:**

This paper introduces Bayesian optimization modeling and optimizing the ranking of evaluations.
To this end, a new surrogate model based on Poisson processes is proposed.
Pairing with the novel ranking-based surrogate model, two acquisition functions are proposed.
The method, PoPBO, outperforms existing methods consistently on simulated functions and hyperparameter optimization benchmarks.
In addition to its superior performance, by replacing Gaussian processes with Poisson processes, it avoids cubic complexity enabling a significant reduction in runtime.

**Summary Of The Review:**

The novelty of the Poisson process surrogate model and its empirical effectiveness with well-paired acquisition functions are the primary contribution of this paper.
Even though it seems that some aspects of the approach are not covered and the presentation of the idea is a bit unclear, I think these can be improved in the rebuttal.
Overall, I think the novel contribution of this paper outweighs its weaknesses.
As long as the presentation of the idea is improved or my concern about the correctness of some argument is resolved, I would increase the score and support for acceptance of this paper.

---

> ### Author Response · Authors · 2022-11-14
> **Reply to Reviewer dFd1 [part 1]**
>
> Thanks for your suggestion, and the point process interpretation is pretty interesting. However, our model is still a discriminative model representing the conditional probability of ranking of candidates $p(\hat{R}_{x}=k|x,\hat{S})$ (Eq. 5).
>
> Also, our method is **not** a single density model since for each point $x$, we assume its ranking as a Poisson distribution whose parameter $\lambda_{\xi}(x)$ is predicted by MLP. Note that there is a typo in Eq. 5 -- $p(\hat{R}_x)$ should be $p(\hat{R}_x|x, \hat{S})$, which we think might mislead your understanding of our method. We have refined it in the new version.
>
> In the following, we would like to first restate the workflow of our PoPBO, which we think will help to understand our methods. Then we will answer your questions one by one. Finally, we would like to state our understanding of your point process interpretation and show its difference from ours.
>
> ### 1. Workflow of PoPBO
> Suppose at $t$-th iteration, we have $N^{(t)}$ history queries, denoted as $\hat{S}^{(t)}$ and their observations. The detailed workflow of our PoPBO at $t$ iteration is as follows (In the following, we omit $t$ without loss of generality.):
>
> 1. **(Get the rankings.)** $\forall x\in \hat{S}$, we compute the ranking of $x$ over $\hat{S}$ by comparing its observation with others and obtain a ranking set $\hat{K} =$ {$\{k_x | \forall x\in \hat{S}\}$}. Similar to [1], we assume the rankings between queries are independent. (Though it is not rigorous, we find it performs better than a truly independent ranking strategy, which will be analyzed later.)
>
>
> 2. **(Compute log-likelihood on observations $\hat{K}$.)** Since the rankings in $\hat{K}$ are independent, we can get the log-likelihood on the observations of ranking $\hat{K}$ as:
> $$
> \begin{equation}
> \log{L(\hat{K}|\hat{S}; \theta)} = \log\Bigg[\prod_{x\in \hat{S}} p(\hat{R}_x=k_x|x, \hat{S})\Bigg] \quad \text{(Independence property)}
> \end{equation}
> $$
>
> $$
> \begin{equation}
> = \sum_{x\in \hat{S}} \log\Bigg[ \frac{\left(\lambda(\xi, x)|X|\right)^k}{k!\cdot Z(x)} \exp{\left(-\lambda(\xi, x)|X|\right)}\Bigg] \text{(According to Eq. 5)}
> \end{equation}
> $$
>
> $$
> \begin{equation}
> =\sum_{x\in \hat{S}}\Bigg[ k_{x} \log{\left(\lambda_\xi(x; \theta)|X| \right)} - \log{(k_x!)} - \log{\bigg[ \sum_{i=0}^{N-1}\frac{\left(\lambda_\xi(x; \theta)|X| \right)^i}{i!} \bigg]} \Bigg] \quad \text{(Derive Eq. 7)}.
> \end{equation}
> $$
>
>
> 3. **(Train the surrogate model.)** We then train MLP by minimizing the log-likelihood through SGD, whose gradient can be computed as Eq. 8.
>
> 4. **(Get the next query.)** We utilize R-LCB or ERI acquisition function to determine the next query. After obtaining the reward, we put the new query and its reward to the observation set $\hat{S}$.
>
>
> **(Independent ranking strategy VS. Our implementation.)**
> Given two points $x_1, x_2$ and a set $S$, their rankings over $S\backslash\{x_1, x_2\}$ are independent. Hence, we implement a truly independent ranking strategy as follows:
> We sample $N/2$ samples from all queries $\hat{S}$ to build a base set $B$. For the left $N/2$ samples, we obtain their rankings over $B$ and build the ranking set $\hat{K} = \{k_x, \forall x \in \hat{S} \backslash B\}$, where $k_x$ denotes the ranking of $x$ over $B$. Such a ranking strategy can guarantee that the rankings in $\hat{K}$ are independent of each other. Then the log-likelihood of the observations of ranking $\hat{K}$ can be computed as:
>
> $$
> \begin{equation}
> \log{L(\hat{K}|\hat{S}\backslash B; \theta)} = \log\Bigg[\prod_{x\in \hat{S}\backslash B} p(\hat{R}_{x}=k_x)\Bigg] \quad \text{(Independence property)}\end{equation}
> $$
>
> $$
> \begin{equation}
> = \sum_{x\in \hat{S}\backslash B} \log\Bigg[\frac{\left(\lambda(\xi, x)|X|\right)^k}{k!\cdot Z(x)} \exp{\left(-\lambda(\xi, x)|X|\right)}\Bigg] \quad \text{(According to Eq. 5)}
> \end{equation}
> $$
>
> $$
> \begin{equation}
> =\sum_{x\in \hat{S}\backslash B}\Bigg[ k_{x} \log{\left(\lambda_\xi(x; \theta)|X| \right)} - \log{(k_x!)} - \log{\bigg[ \sum_{i=0}^{|B|-1}\frac{\left(\lambda_\xi(x; \theta)|X| \right)^i}{i!} \bigg]} \Bigg] \quad \text{(Derive Eq. 7)}.
> \end{equation}
> $$
>
> However, the above independent ranking strategy results in under-utilization of observations $\hat{S}$ since it has to be divided into two sets. Moreover, candidates with different performances may get the same ranking over $B$, which will mislead the training of the surrogate. This work adopts a simple relaxation -- We compute $\hat{R}$ as the ranking over the whole set $\hat{S}$ and assume independence property, which is also utilized in [1]. Our experimental results show that such a relaxation results in better performance since it can fully utilize all observations.
>
> [1] David Salinas, Huibin Shen, and Valerio Perrone. A quantile-based approach for hyperparameter transfer learning. In ICML, 2020.

---

> > ### Author Response · Authors · 2022-11-14
> > **Reply to Reviewer dFd1 [part 2]**
> >
> > ### 2. Response to the Questions
> >
> > * ### 2.1 The explanation of the Poisson process surrogate model
> >
> > **Q1: In Eq.(2), does the definition of $\hat{R}_x(S)$ depend on the choice of $\hat{S}$? This does not seem a natural way to define Poisson processes.**
> >
> > A1:
> > $\hat{R}_{x}(S)$ depends on the choice of $\hat{S}$.
> >
> > We would like to show that $\hat{R}_{x}(S)$ can obey the two properties of the Poisson process.
> >
> > On the one hand, $R_x(\emptyset) = 0$ since $\emptyset\cap S = \emptyset, \forall S\subset X$. On the other hand, $\hat{S}$ contains discrete points, so $\forall S\subset X$, we can find a small enough $\Delta s$ satisfying that $\lim_{\Delta s\rightarrow0}\text{P}(\hat{R}(S+\Delta s|x) - \hat{R}(S|x)\geq2) = 0$.
> >
> > **Q2: Why $\delta$ is necessary? What is the rigorous definition of 'discrete'?**
> >
> > A2: We would like to first give the definition of continuous, and then define discrete through the negative continuous proposition.
> >
> > *Definition of continuous.* If $\forall \mu>0$, $\exists \delta>0$, $\forall x'\in \delta(x)\triangleq \{x':|x'-x|<\delta\}$ satisfying $|f(x')-f(x)|<\mu$, we say $f(x)$ is continuous at point $x$.
> >
> > *Definition of discrete (negative continuous proposition).* If $\exists \mu>0$, $\forall \delta>0$, $\exists x'\in \delta(x)$ satisfying $|f(x')-f(x)|\geq\mu$, we say $f(x)$ is discrete (not continuous) at point $x$.
> >
> > Consider the proof of proposition 1, we show that $\forall \delta>0, \forall x': |x'-x|<\delta, |f(x')-f(x)|=|\hat{f}(x')-\hat{f}(x)+\Delta\epsilon|$, where $\Delta\epsilon\sim\mathcal{N}(0, 2\sigma^2)$. Hence, $P(\exists x'\in\delta(x), |f(x')-f(x)|\geq\mu) =1-P(\forall x'\in\delta(x), |f(x')-f(x)|<\mu) =$ $1-\prod_{x'\in\delta(x)}P(\Delta\epsilon\in(-\mu-\hat{f}(x')+\hat{f}(x), \mu+\hat{f}(x')-\hat{f}(x))) \approx 1$, since $f(x)$ is continuous in $\delta(x)$. Therefore, we can say that $f(x)$ is almost surely discrete anywhere. We have put the above analysis in Sec. A in the supplementary.
> >
> > **Q3: How to conclude the property 2)?**
> >
> > A3: For a continuous black-box function, it is a strong assumption that the ranking of $x$ over the whole set $S$ satisfies property 2, even if we derive proposition 1. However, in our implementation, $\hat{R}_x(S)$ depends on a discrete set $S\cap\hat{S}$, which actually discretizes the continuous space $S$. Hence, we can find a small enough $\Delta s$ satisfying $\Delta s\cap \hat{S}=\emptyset$. Therefore, property 2 (sparse assumption) is naturally satisfied in our implementation.
> >
> > * ### 2.2 Confusing notations
> >
> > **1.** Thanks for pointing out the typo. In the 4-th line of Sec. 3.2, $N(S|x)$ should be $\hat{R}_x(S)$. For conciseness, we use $\hat{R}_x(S)$ to denote the ranking of $x$ over a set $S\cap\hat{S}$ as shown in Eq. 2.
> >
> > **2.** $|\hat{S}|$ is the number of elements of $\hat{S}$ and $|X|$ is the volume of $X$.
> >
> > **3.** Thanks for pointing out the typo. In Eq.(2), it should be $|S\cap \hat{S}|$ without curly bracket.
> >
> > **4.** Given a point x, for a specific set $S$, $\hat{R}_x(S)$ is a Poisson random variable.
> >
> > In the original version, we utilize $\mathcal{P}_{nh}$ to denote a non-homogeneous Poisson process, which should be denoted as $\hat{R}_x(S) \sim Poisson(\int_S \lambda(s,x)\text{d}s)$. We refined it in the new version.

---

> > > ### Author Response · Authors · 2022-11-14
> > > **Reply to Reviewer dFd1 [part 3]**
> > >
> > > * ### 2.3 How good is the uncertainty of the Poisson process surrogate model?
> > >
> > > **Q1: It seems that the trick 'rectification' is crucial in performance, which may imply that the Poisson process surrogate model may not be good as expected.**
> > >
> > > A1: We conduct ablation studies on the effect of rectification in Fig. 6. No rectification is applied to PoPBO when $q=0$. Comparing the results of GP-BO (Fig. 2) and PoPBO without rectification ($q=0$ in Fig. 6), we observe that PoPBO without rectification still outperforms GP, showing the advantage of ranking response surface.
> > >
> > >
> > > **Q2: the Poisson process surrogate model is a single density model fitted with MLE, so the weak uncertainty may come from this training procedure.**
> > >
> > > A2: We would like to argue that our surrogate model considers the uncertainty. In contrast, the poor uncertainty mainly lies in the unsuitable vanilla acquisition functions that are designed for GP, which motivates us to design R-LCB and ERI (as mentioned in the third line in Sec. 4.). Our explanations are as follows:
> > >
> > > **1. How does PP consider the uncertainty?**
> > > First of all, we would like to point out that PoPBO is **not** a single density model. $\forall x\in X$, the ranking of $x$ obeys a Poisson distribution $\text{P}(R_x|x)$ shown as Eq. 5, and the ranking of different points obeys different distributions.
> > > Hence, the uncertainty of the ranking of every point has been considered. Specifically, according to the property of Poisson distribution, the variance of ranking of each point is the same as its mean, controlled by $\lambda_\xi(x)$. This work utilizes an MLP to predict $\lambda_\xi(x)$.
> > > Hence, PP considers the uncertainty of ranking of each point and the training procedure bears little responsibility for the weak uncertainty.
> > >
> > >
> > > **2. Why are the vanilla acquisition functions unsuitable for PP resulting in poor uncertainty?**
> > > Gaussian distribution has independent mean and variance. The vanilla acquisition functions, designed for GP, take independent mean and variance as inputs, and the point with high variance is also probable to be selected even if its mean is very low. In contrast, the variance of Poisson distribution is the same as its mean, indicating that a point with a low mean value has a low variance. Therefore, directly applying the vanilla acquisition functions is unsuitable, since minimizing the acquisition function will select the point with the smallest mean and variance, which will harm the uncertainty leading to an over-exploitation issue (as analyzed in the paragraph below Eq. 10 in the original version). Therefore, we propose the rectification technique for better exploration-exploitation balance.
> > >
> > > * ### 2.4 Point Process Interpretation
> > >
> > > **Q1: How do we model the probability of a set $\{ x_1, x_2, \cdots, x_N \}$ from PP model?**
> > >
> > > A1: Please see our detailed introduction to the PoPBO workflow. Since we assume the observations of rankings are independent, we can compute the joint log-likelihood as Eq. 7.
> > >
> > > **Q2: $\hat{R}_x(S)$ is defined on $\hat{S}$, which is sampled randomly.**
> > >
> > > A2: We would like to first restate the notations in our paper to resolve your concern.
> > >
> > > 1. $X$ is the whole feasible domain (search space).
> > > 2. $S_x = \{y|y\in S, f(y)<f(x)\}$ is the set of better points than $x$ in $S\subset X$, where $S$ can be any continuous domain or discrete set.
> > > 3. $\hat{S}$ is a discrete set containing both initial samples for BO and the history queries.
> > > 4. Given a specific $S$, $\hat{R}_x(S)$ is a random variable denoting the possible ranking of $x$ over a discrete set $\hat{S}_x\cap\hat{S}$. Hence, $\hat{R}_x(S)$ depends on $\hat{S}$, we utilize a hat symbol $\hat{}$ to simplify the notation.
> > >
> > > Moreover, the new queries and the observations will be put into $\hat{S}$ during the BO run, so our $\hat{R}_x{S}$ is defined as the number of random points that are a set of queries generated by PoPBO.
> > >
> > >
> > > **Q3: What is the workflow of PoPBO when $N$ evaluations are given.**
> > >
> > > A3: Please refer to our detailed introduction of the workflow of PoPBO. When $N$ evaluations are given, we first get the rankings of all observations, and then compute log-likelihood based on Eq. 7. Next, we train the surrogate model (MLP) by minimizing the log-likelihood. Finally, the next query is obtained by minimizing the acquisition function.

---

> > > > ### Author Response · Authors · 2022-11-14
> > > > **Reply to Reviewer dFd1 [part 4]**
> > > >
> > > > ### 3. Our Understanding of Point Process Interpretation
> > > >
> > > > Last but not the least, we would like to state our understanding of your point process interpretation. Please do not hesitate to reply if there is any misunderstanding.
> > > >
> > > > First of all, we would like to state that PoPBO is **not** a single density model. PoPBO assumes that $\forall x\in X$, the ranking of it obeys a specific Poisson distribution $\text{P}(R_x|x)$ shown as Eq. 5. From this point, we think PoPBO can be seen as one of the implementations of your point process interpretation.
> > > >
> > > > Second, the log-likelihood is computed as Eq. 7 based on an independent assumption, which is similar to [1]. We also implement another ranking strategy leading to truly independent among rankings, whose performance is slightly worse due to the under-utilization of observations. It can be seen as a limitation of our PoPBO, which we would like to leave as future work.
> > > >
> > > > Third, we are pretty interested in your brand-new idea about modeling the sequence order. However, with the process of BO run, more points are queried with observations, resulting in different conditional probability, which is hard to be established by a unified model.
> > > >
> > > > Finally, we would like to state that PoPBO is a discriminative model rather than a generative model since Eq. 9 is a {\color{red} conditional probability} of the ranking of a query $x^*$.
> > > >
> > > > [1] David Salinas, Huibin Shen, and Valerio Perrone. A quantile-based approach for hyperparameter transfer learning. In ICML, 2020.

---

> > > > > ### Comment · Reviewer_dFd1 · 2022-11-29
> > > > > **Response to the rebuttal**
> > > > >
> > > > > Thanks for the answers.
> > > > >
> > > > > - 'Single' model. It seems that my usage of ‘single’ confused the authors. I understand that at each point it has different Poisson distributions. What I intended is to contrast the way PP is fitted with a fully Bayesian treatment of GP hyperparameter fitting (which is basically the average of models with different hyperparameters).
> > > > >
> > > > > - The way the PP is fitted is it assumes a parametric model approximating Poisson distributions at all points in the search space in an amortized way, since one parametric model is shared to model multiple Poisson distributions at all points. I think in this way somehow dependence among Poisson distributions at different points is implicitly modeled. It looks unintuitive that Poisson distributions at different points are independent. It seems that such independence cannot be abandoned because factorization allows tractable density computation. With such independence assumption, it is quite difficult to understand what the PP is modeling. If all rankings are independent, then the surrogate model cannot do anything since it relies on the fact that somehow the information changes less in neighboring regions.
> > > > >
> > > > > - It seems that PP fits is a conditional probabilistic model, i.e. Poisson distribution conditioned at some location in the search space. Then this is quite different from how GP surrogate model works in which the posterior is used being conditioned by given observations. By doing MLE fitting, it is unclear how the uncertainty is properly handled. Even though the authors argue that the PP produces a reasonable uncertainty but such good uncertainty is poorly handled in existing acquisition functions. This argument requires some reasonable empirical evidence or mathematical rigor.
> > > > >
> > > > > - It seems that the assumptions for Poisson process are satisfied. And it seems OK to consider it granted. However, the argument for property 2 is still confusing to follow.
> > > > >
> > > > > Most of all, it is not clear what PP is modeling especially with the authors' argument that Poisson distributions are independent. Also. it lacks an analysis of the uncertainty of PP in spite of the MLE fitting of PP, which can generate poor uncertainty maybe due to overfitting.
> > > > > Due to such concerns, I keep my score.

---

### Official Review · Reviewer_v6dz · 2022-10-24

**Confidence:** 5
**Correctness:** 3
**Technical Novelty And Significance:** 2
**Empirical Novelty And Significance:** 2
**Recommendation:** 5

**Clarity, Quality, Novelty And Reproducibility:**

The paper is well-written, and I enjoyed reading it. I am happy to change my scores per the rebuttal and the execution of the extra experiments.

**Strength And Weaknesses:**

Strengths:
- Well-written paper
- Clear set of contributions
Weaknesses:
- Experimental study: Although the authors perform an extensive experimental study, I am left wondering why HEBO (heteroscedastic and evolutionary Bayesian optimisation) is not a baseline? The reason I ask is that such an approach introduces input and output warping functions to `correct' the noise process of the observed data. It also changes the acquisition to a multi-objective one. It has also been demonstrated to outperform some of the baselines presented in the paper. I would like to kindly ask the authors to baseline and compare against newer SOTA methods (HEBO being one of them).
- The authors demonstrate that their techniques enjoy lower computational complexity of O(N^2) compared to the GP O(N^3). This is interesting and worth emphasising. However, how does this compare to induced point GPs? Can the authors perform scalability experiments compared to variational GPs for instance? What would the cost look like in this case?
-  In Figure 2, it seems that on 2d Branin and 6d Hartmann, GP-EI performs the best. If that is the case, when should we use the algorithm proposed by this paper versus standard BO? Can the authors help give guidance on the applicability domains of their technique?
- The same question goes for when to use R-LCB versus ERI. Can the authors elaborate on those results in accordance with Table 1?



**Summary Of The Paper:**

In this paper, the authors propose a method for Bayesian optimisation based on Poisson processes. The interesting concept is that the authors introduce novel ranking response surfaces in addition to two acquisition functions that exploit the proposed model. They perform experiments on various benchmarks and demonstrate favourable results compared to the baseline technique.

The paper is well-written, easy to follow, and nice to read.


**Summary Of The Review:**

Please see above.

---

> ### Author Response · Authors · 2022-11-14
> **Reply to Reviewer v6dz**
>
> **Q1: More baselines in the experimental study.**
>
> A1: Thanks for your kind suggestion. We add the comparison with HEBO on Hartmann and Rosenbrock simulation functions in Fig. 2.
>
> **Q2: More comparison of computational complexity to induced point GPs and variational GPs.**
>
> A2: The computational complexity of induced point GP is $O(NM^2)$, where M is the number of induced points and is usually set as $1/4N$. In general, induced point GP still has larger computational complexity compared to PoPBO with $O(N^2)$ computational complexity.
> Moreover, we utilize the implementation of variational GP in BoTorch (https://github.com/pytorch/botorch/blob/main/botorch/models/approximate_gp.py) and test its time cost on the 6d-Hartmann benchmark. We report the time cost of PoPBO and induced point GP in the following table, which verifies the above analysis. All methods are run on an Intel(R) Xeon(R) Silver 4210R CPU.
>
> |                  | 100   | 200    | 300     | 400     | 500   |
> | ---------------- | ----- | ------ | ------- | ------- | ----- |
> | PoPBO            | 37.6 s  | 111.9 s | 225.6 s  | 385.2  s | 575.2 s |
> | Induced Point GP | 907.5 s | 7202.5  s | 11641.6 s | 14151.9 s | 19642.1   s   |
>
>
>
> **Q3: In Figure 2, it seems that on 2d Branin and 6d Hartmann, GP-EI performs the best. If that is the case, when should we use the algorithm proposed by this paper versus standard BO? Can the authors help give guidance on the applicability domains of their technique?**
>
> A3: To verify the robustness to noise of PoPBO, we add Gaussian noises with zero mean and various standard deviations ($\sigma$) to Hartmann and run GP-BO and PoPBO separately. Results are reported in Fig. 11 (supplementary) and analyzed in Sec. C.7. Specifically, we observe that PoPBO outperforms GP when the objective function has large noise ($\sigma=0.1$), demonstrating the robustness of PoPBO to noise.
>
> **Q4: When to use R-LCB versus ERI. Can the authors elaborate on those results in accordance with Table 1?**
>
> A4: It is a pretty good but tough question, however, we are afraid that it is still inconclusive in the academic field that which acquisition function performs better, even for GP-BO which has been explored for years. The results of NAS-Bench-201 (Table 1) show a minor gap between ERI and R-LCB, making it hard to distinguish which one is better. While in Fig. 2c (6-d Rosebrock), we observe that R-LCB can be more suitable for large-space problems. This work mainly aims to introduce a brand-new ranking response surface, and derive ERI and R-LCB from two of the most popular acquisition functions (EI and LCB) to accommodate our response surface. We would like to leave the exploration of acquisition functions to our future work.

---

> > ### Comment · Reviewer_v6dz · 2022-12-11
> > **Thank you**
> >
> > Thanks for the authors for the feedback. I have read thoroughly through it. While some extra experiments have been added, I still believe some answers are not clear yet. This is especially true on understanding the empirical results. I asked questions related to applicability of the method but got responses on robustness. Maybe the authors can help me understand how this sheds light on my original question?

---

### Official Review · Reviewer_rWSY · 2022-10-28

**Confidence:** 4
**Correctness:** 2
**Technical Novelty And Significance:** 3
**Empirical Novelty And Significance:** 2
**Recommendation:** 3

**Clarity, Quality, Novelty And Reproducibility:**

As described above, this paper is tough to follow since several critical pieces of information were omitted. Moreover, the code to reproduce the experiments was not included. Thus, this work performs very poorly in terms of clarity and reproducibility. The proposed modeling approach is novel and technically sound. On the other hand, the proposed acquisition functions seem too heuristic and hard to trust in practice. Overall, I believe this paper enjoys significant novelty but low execution quality.

**Strength And Weaknesses:**

Strengths:
1. The proposed modeling approach is novel and technically sound.
2. The empirical performance of the proposed approach is promising.

Weaknesses:
1. The problem setting and proposed probabilistic model descriptions are hard to follow. For example, it is not specified what the observations are. This is more clearly articulated in the supplement, where the ranking function is mentioned. The description of the probabilistic model is also hard to follow since MLP and $\theta$ are not defined.
2. The proposed acquisition functions are too heuristic. They both rely on hyperparameters which seem to affect their performance significantly. It would be more appealing to derive an acquisition function based on an information-theoretic or decision-theoretic analysis.
3. Some aspects of the empirical evaluation were not specified. For example, was the domain discretized? If not, how is $|X|$ handled in equations 6-9? Also, was noise added to the objective function observations in the numerical experiments? If so, this should be stated. Moreover, the effect of the noise level should be assessed to support the claim that the proposed approach is more robust to noise than standard BO approaches.
4. The proposed modeling approach is technically sound overall. However, the independence of rankings over disjoint sets seems counterintuitive. In general, one would expect the rankings of nearby regions to be correlated.

**Summary Of The Paper:**

This work proposes a novel modeling approach within the Bayesian optimization (BO) framework. More concretely, it proposes to use a (truncated) Poisson process to model the ranking induced by the underlying objective function over a feasible domain. It is argued that such an approach is more robust to noise in the objective function observations. Two acquisition functions inspired by the GP-UCB and EI acquisition functions are proposed. The performances of the proposed approach and several benchmarks from the literature are compared across several synthetic and realistic test problems, showing favorable results.

**Summary Of The Review:**

This work proposes a novel probabilistic model within the Bayesian optimization framework. While I have some concerns about the proposed approach, it is technically sound overall, and its empirical performance is promising. Unfortunately, this paper is tough to follow, with several missing details and poor descriptions. In addition, the proposed acquisition functions, which would play a significant role in making the proposed modeling approach successful in practice, seem too heuristic and complicated to use due to their dependence on hyperparameters. Overall, I believe this work requires significant improvements to merit publication at a venue such as ICLR.

---

> ### Author Response · Authors · 2022-11-14
> **Reply to Reviewer rWSY**
>
> **Q1: It is not specified what the observations are. The description of the probabilistic model is also hard to follow since MLP and $\theta$ are not defined**
>
> A1: Observations are the rankings of each query among a set $\hat{S}$, which is defined in Eq. 2 and mentioned in the paragraph above Eq. 7 in the main text (in the original version). The architecture of Multi-layer perception (MLP) and the training settings are also introduced in Sec. 5 (Lines 7-9 on page 7).
>
> **Q2: The proposed acquisition functions are too heuristic. They both rely on hyperparameters which seem to affect their performance significantly. It would be more appealing to derive an acquisition function based on an information-theoretic or decision-theoretic analysis.**
>
> A2: Thanks for the kind suggestion, however, the main contribution of PoPBO aims at a novel relative response surface based on the Poisson process. The acquisition functions are derived from LCB and EI to fit our response surface. We believe that our work provides a brand-new perspective to directly capture the rankings of candidates, in the hope of shedding light on this topic. We would like to leave the research on new acquisition functions as our future work.
>
> **Q3: Was the domain discretized and how is $|X|$ handled in equations 6-9?**
>
> A3: PoPBO supports either continuous or discrete feasible domains. If $X$ is continuous, $|X|$ is the volume of $X$.
>
> **Q4: Was noise added to the objective function observations in the numerical experiments? The effect of the noise level should be assessed to support the claim that the proposed approach is more robust to noise than standard BO approaches.**
>
> A4: The observations in the simulation function are noiseless, while the observations in the real-world benchmark (HPO-Bench and NAS-Bench-201) are noisy.
> For HPO-Bench, the performance of each configuration (hyperparameters of FCNet) is evaluated 4 times under different random seeds. In the experiment settings of HPO-Bench, the average performance is used as the observation, which is naturally noisy.
> NAS-Bench-201 attempts to search for a neural architecture that performs best after 200 training epochs. However, following the experiment settings of NAS-Bench-201, only the validation accuracy after 12 training epochs of each architecture can be queried, making the observations noisy.
> Moreover, We conduct experiments on noisy simulation functions with various noise levels, whose results are shown in Fig. 11 (supplementary) and analyzed in Sec. C.7. Specifically, we observe that PoPBO outperforms GP when the objective function has large noise ($\sigma=0.1$), demonstrating the robustness of PoPBO to noise.
>
> **Q5: The independence of rankings over disjoint sets seems counterintuitive. In general, one would expect the rankings of nearby regions to be correlated.**
>
> A5: Here, we take $y=x^2$ as a simple example to show that the ranking of the same point over nearby sets can be different and depends on the position of the sets. Consider a point $x=2$, its ranking over two nearby sets {$2.1, 2.2, \cdots, 2.9$} and {$1.1, \cdots, 1.9$} can be quite different (1 vs. 10). However, its ranking over another two nearby sets {$10.1, 10.2, \cdots, 10.9$} and {$11.1, 11.2, \cdots, 11.9$} are the same. There will be more such situations when $f(x)$ has many extremum points, such as $sin(x)$. In this work, since $f(x)$ is a black-box objective function and we have no prior on it, hence, we assume the rankings over two disjoint areas are independent.
>
> For the case where $f(x)$ is not black-box and the rankings over disjoint sets are dependent, the ranking response can be modeled by the conditional Poisson process, which we think is beyond the scope of this paper and we would like to explore it in the future.

---

> > ### Comment · Reviewer_rWSY · 2022-11-14
> > **Reviewer rWSY's post-rebuttal follow-up**
> >
> > Dear authors,
> >
> > Thank you for your responses. Please find follow-up questions/comments below:
> >
> > Q1 follow-up: The recently introduced notational changes help clarify what the training data is. My concern now is that $\hat{R}_x(S)$ is no longer being defined in Eq. 2. Was this change intentional? Separately, I want to emphasize that the definition of the acronym MLP is still absent in the new version of the paper.
> >
> > Q2 follow-up: While I understand that the main contribution of this work is the proposed probabilistic model, the usefulness of such a model in the context of BO depends on the ability to derive an acquisition function that can be trusted in practice. At the same time, I want to clarify that I am not expecting the authors to introduce a new acquisition function during the rebuttal period.
> >
> > Q3 follow-up: It should be clarified that $|X|$ denotes the cardinality of $X$ if $X$ is finite and its volume if $X$ is continuous.
> >
> > Q4 follow-up: Thank you for this clarification. Please include this information in the appendix. It would be desirable to conduct similar experiments for the other synthetic test functions.
> >
> > Q5 follow-up: These examples do not agree with what I had in mind as nearby regions. Consider the sets $\{1, 2, 3\}$ and $\{1 + \epsilon, 2 + \epsilon, 3 + \epsilon\}$. For $\epsilon < 1$, these sets are disjoint, and thus the rankings of x over these two sets are independent. However, one would expect these rankings to be highly correlated for $\epsilon$ small enough.
> >
> > Finally, I would like the authors to comment on the possibility of releasing their code upon acceptance. Given that the proposed model is non-standard, I believe this is crucial to guarantee that the experiments in this paper can be easily reproduced.

---

> > > ### Author Response · Authors · 2022-11-14
> > > **Reply to Reviewer rWSY**
> > >
> > > * **Q1 follow-up: $\hat{R}_x(S)$ is no longer being defined in Eq. 2. Was this change intentional? The definition of the acronym MLP is still absent in the new version of the paper.**
> > >
> > > A1:
> > > 1) We found that the definition of $\hat{R}_x(S)$ in Eq. 2 and Eq. 4 in the original version can be a little confusing, so we clarify that $\hat{R}_x(S)$ denotes a random process in our new version. Please see our revision in the first paragraph in Sec. 3.2.
> > > 2) Thanks for the reminder, we specify the definition of MLP in this version (in the third line below Eq. 6 marked in blue).
> > >
> > > * **Q2 follow-up: While I understand that the main contribution of this work is the proposed probabilistic model, the usefulness of such a model in the context of BO depends on the ability to derive an acquisition function that can be trusted in practice. At the same time, I want to clarify that I am not expecting the authors to introduce a new acquisition function during the rebuttal period.**
> > >
> > > A2: We are not aiming to provide a new acquisition function during the rebuttal period. In contrast, our proposed acquisition functions apply for the Poisson response surface in practice, which has been proved by the extensive experimental results. Therefore, the heuristic but effective R-LCB and ERI can be treated as one of the contributions of this work.
> > >
> > >
> > > * **Q3 follow-up: It should be clarified that $|X|$ denotes the cardinality of X if X is finite and its volume if X is continuous.**
> > >
> > > A3: Thanks for the reminder, and we clarified it in Sec. D in the new version.
> > >
> > > * **Q4 follow-up: Please include the information of noise in the appendix. It would be desirable to conduct similar experiments for the other synthetic test functions.**
> > >
> > > A4: 1) We add the information of noise to Sec. C.7 in the new version. 2) The experimental results of Hartmann with various noises have been reported and analyzed in Fig. 11 and Sec. C.7. Other experiments on noisy Rosenbrock and Branin are running and we will update the result by the rebuttal deadline.
> > >
> > > * **Q5 follow-up: These examples do not agree with what I had in mind as nearby regions. Consider the sets {$1, 2, 3$} and {$1+\epsilon, 2+\epsilon, 3+\epsilon$}. For $\epsilon < 1$, these sets are disjoint, thus the rankings of x over these two sets are independent. However, one would expect these rankings to be highly correlated for $\epsilon$ small enough.**
> > >
> > > A5: The rankings of $x$ over the two sets are independent even for a small enough $\epsilon$. The reasons are as follows:
> > >
> > > Suppose, $f(x)$ is an observation of a black-box function with independent additive noise, i.e. $f(x) = \hat{f}(x)+\mu$, where $\mu$ denotes noise. Given a reference point $x$ with observation $f(x)$ and any other two points $x_1, x_2$ with black-box function values $\hat{f}(x_1), \hat{f}(x_2)$. We define the ranking of x over a single set {$x_1$} as random variable $r_1$, which obeys a Bernoulli distribution:
> > > $$P(r_1=1) = P(\mu_1 < f(x)-\hat{f}(x_1)); \quad P(r_1=2) = P(\mu_1 >= f(x)-\hat{f}(x_1))$$
> > > Similarly the ranking of x over another single set {$x_2$}, denoted as $r_2$ also follows:
> > > $$P(r_2=1) = P(\mu_2 < f(x)-\hat{f}(x_2)); \quad P(r_2=2) = P(\mu_2 >= f(x)-\hat{f}(x_2))$$
> > > Since $\mu_1$ and $\mu_2$ are noises independent from each other, we have $P(r_1, r_2)=P(r_1)*P(r_2)$.
> > >
> > > We simplify your setting as two sets {$x_1$} and {$x_2:x_2=x_1+\epsilon$}, where $\epsilon$ is small enough. Even if $\hat{f}(x_1)$ = $\hat{f}(x_2)$, since $\mu_1$ and $\mu_2$ are independent, $P(r_1=1, r_2=1)=P(u_1<f(x)-\hat{f}(x_1)，u_2<f(x)-\hat{f}(x_1))=$$P(u_1<f(x)-\hat{f}(x_1))*P(u_2<f(x)-\hat{f}(x_1))$. Hence, the independent property holds.
> > >
> > > The above property is also satisfied for multi-point cases, such as {$1,2,3$} and {$1+\epsilon, 2+\epsilon, 3+\epsilon$}. Therefore, even if the black-box function is continuous, the independent increment assumption is rational.
> > >
> > >
> > > * Finally, we will definitely open-source our codes upon acceptance.

---

> > > > ### Comment · Reviewer_rWSY · 2022-11-23
> > > > **Reviewer rWSY's post-rebuttal update**
> > > >
> > > > Dear authors,
> > > >
> > > > I appreciate your response. Some of my concerns have been addressed. However, I am still worried about the proposed acquisition functions. Moreover, I am now even more concerned about the assumption of the independence of rankings over disjoint sets. I believe $\hat{R}_x(S)$ is intended to model the ranking induced by the noise-free function (which in your example you are referring to as $\hat{f}$) using noisy observations, as opposed to the ranking induced by the noisy observations directly ($f$). If this is the case, then your example is invalid. If this is not the case, then I believe your approach is fundamentally wrong, as this means it is aiming to find the optimum of $f$ instead of $\hat{f}$.
> > > >
> > > > Given these concerns, I have decided to keep my score.
> > > >
> > > > Best wishes,
> > > > Reviewer rWSY

---

### Author Response · Authors · 2022-11-14
**Revision Details**

We thank all the reviewers for their valuable comments. We refine the manuscript and supplementary marked in blue according to the suggestions.

### Rectification of Notations
1. For any fixed $S\subset X$, $\hat{R}_x(S)$ is a random variable denoting the possible ranking of $x$ over $S\cap\hat{S}$.
2. We utilize $k_x$ to denote an observation of ranking $x$ over the feasible domain $X$. $K=\{k_x\}$ consists of the observed rankings of all samples.

3. Since $\hat{R}_x(S)$ depends on $\hat{S}$ and $x$, we rectify the notation of probability in Eq. 5 and Eq. 9.


### Revision of the Writing
1. We specify the 'Efficient' in the third contribution as 'Computational-efficient' (as suggested by Reviewer 55na).
2. We add footnotes to define 'absolute response' and 'relative response' (as suggested by Reviewer 55na).
3. We revise the proof of Proposition 1 in the supplementary (as suggested by Reviewer dFd1).

### New Experiments

1. We add the comparison with HEBO in Fig. 2. (Required by Reviewer v6dz.)
2. Comparison of time cost with variational GP. (Required by Reviewer v6dz.)
3. We implement GP-MLP by utilizing an MLP to estimate the parameters of the Gaussian process. Results are reported in Fig. 8 and Sec. C.4. (Required by Reviewer 55na.)
4. We add the ablation study on the worst tolerant ranking $K_m$ for ERI. Results are reported in Fig. 9 and Sec. C.5. (Required by Reviewer 55na.)
5. We run GP-BO and PoPBO on 6-d Rosenbrock for 200 iterations. Results are reported in Fig. 10 and Sec. C.6. (Required by Reviewers 55na.)
6. We compare the performance of PoPBO and GP on various noise levels on simulated benchmarks. Results are reported in Fig. 11 and Sec. C.7. (Required by Reviewers rWSY, v6dz, and 55na.)

---

> ### Author Response · Authors · 2022-11-17
> **New Revision**
>
> We updated the experimental results of the robustness to various noise levels in Fig. 11. Specifically, we conducted experiments on Branin and Rosenbrock simulated benchmarks. Please feel free to ask us any questions if you have concerns about our work.

---

### Decision · Program_Chairs · 2023-01-20

**Decision:**

Reject

**Justification For Why Not Higher Score:**

- Concern about rigor
- Lack of justification for proposed approach

**Justification For Why Not Lower Score:**

N/A

**Metareview: Summary, Strengths And Weaknesses:**

This paper considers Bayesian optimization where we observe rankings over points' rewards, rather than the rewards directly. This has the potential to be more robust to noise. The paper proposes a truncated Poisson process surrogate and two novel acquisition functions. The proposed method is evaluated on several synthetic and real-world problems.

Strengths
- The idea of using rankings rather than absolute observations seems valuable
- Novel approach
- Evaluation shows consistent improvement over existing methods
- New method offers reduced runtime over standard Gaussian processes

Weaknesses
- Proposed acquisition functions lack justification and are sensitive to their hyperparameters
- Substantial lack of clarity, both in terms of the problem setting, model, and experimental evaluation.
- The reviewers raised several concerns about the rigor of the analysis: dFd1's first and second weakness; the concerns about the paper's rigor from 55na, and rWSY's concern that rankings over disjoint sets cannot be treated as indpendent. Some of these may simply be due to a lack of clarity and small errors in mathematical descriptions, but the concern about independence of rankings over disjoint sets remains substantial following a detailed and careful discussion between rWSY and the authors.